# Tempo: an unsupervised Bayesian algorithm for circadian phase inference in single-cell transcriptomics

Benjamin J. Auerbach [1] ✉, Garret A. FitzGerald[2] & Mingyao Li [3] ✉

The circadian clock is a 24 h cellular timekeeping mechanism that regulates human physiology. Answering several fundamental questions in circadian biology will require joint measures of single-cell circadian phases and transcriptomes. However, no widespread experimental approaches exist for this purpose. While computational approaches exist to infer cell phase directly from single-cell RNA-sequencing data, existing methods yield poor circadian phase estimates, and do not quantify estimation uncertainty, which is essential for interpretation of results from very sparse single-cell RNA-sequencing data. To address these unmet needs, we introduce Tempo, a Bayesian variational inference approach that incorporates domain knowledge of the clock and quantifies phase estimation uncertainty. Through simulations and analyses of real data, we demonstrate that Tempo yields more accurate estimates of circadian phase than existing methods and provides well-calibrated uncertainty quantifications. Tempo will facilitate large-scale studies of single-cell circadian transcription.

The circadian molecular clock is a 24 h timekeeping mechanism found in nearly every cell in humans[1]. The time of the clock, referred to as circadian phase, is determined by the mRNA and protein concentrations of the clock's constituent genes, referred to as clock or core clock genes[2]. Clock genes are organized in a transcriptional-translational feedback loop that enables cells to maintain self-sustained ~24 h oscillations in the concentrations of clock gene mRNA. Clock gene proteins additionally interact with cell-type-specific regulatory factors to drive rhythmic transcription of genes referred to as clock-controlled genes (CCGs). It is partially through these CCGs that circadian clocks generate rhythmic cellular behaviors, such as rhythms in hepatocyte glycogenesis[3] and vascular smooth muscle cell (SMC) contractility[4]. Though self-sustained, circadian clocks additionally rely on environmental cues, referred to as Zeitgebers (e.g., light), to update and optimize their timing via a process referred to as entrainment[5].

Many open questions in chronobiology require single-cell resolution, such as the identification of cell-type-specific CCGs and the role of circadian phase in gating cell fate decisions. As droplet-based single-cell RNA-sequencing (scRNA-seq) measures genome-wide single-cell transcriptomes at high throughput, it has become an attractive tool with which to study many of these questions. Existing scRNA-seq studies of the clock have relied on time course designs[6–8], in which cell clocks are presumed to be entrained by an external rhythmic stimulus, such as light. Assuming cell clocks are perfectly synchronized, sample timing over the cycle of the stimulus can be used as a direct proxy for the circadian phases of all cells in the sample. Nevertheless, this is a limiting assumption, as previous studies suggest cell circadian phases can differ by several hours within the same tissue in vivo and are determined by biological variables such as spatial location[9–12]. Furthermore, chronobiologists may be interested in studying circadian transcriptional rhythms of cells in the absence of timing cues

[1]Graduate Group in Genomics and Computational Biology, University of Pennsylvania Perelman School of Medicine, Philadelphia, PA 19104, USA. [2]Institute for Translational Medicine and Therapeutics, University of Pennsylvania Perelman School of Medicine, Philadelphia, PA 19104, USA. [3]Department of Biostatistics, Epidemiology and Informatics, University of Pennsylvania Perelman School of Medicine, Philadelphia, PA 19104, USA. ✉e-mail: benauer@pennmedicine.upenn.edu; mingyao@pennmedicine.upenn.edu

(e.g., unsynchronized cells in a dish). Breaking this assumption requires single-cell measures of circadian phase. One approach is to estimate cell circadian phases from gene expression directly, a task referred to as unsupervised phase inference.

Several algorithms have been developed for the similar task of unsupervised phase inference for cell cycle analysis using scRNA-seq data[13–16]. However, the circadian cycle and cell cycle differ in two notable ways. First, while hundreds of "core" genes are known to oscillate over the cell cycle, many of which are highly expressed[17,18], the core circadian clock is only comprised of ~20 moderately expressed genes[2]. Second, ~100–1000 CCGs[6–8] oscillate in a cell-type-specific manner over the circadian cycle and the identities of these genes are often unknown ahead of time. Due to the moderate expression of circadian clock genes and the challenge in identifying CCGs, existing unsupervised phase inference methods perform poorly when tasked with ordering cells over the circadian cycle. An optimal approach for estimating circadian phase in scRNA-seq should thus identify CCGs de novo and incorporate their information into phase estimates.

Existing unsupervised phase inference approaches were mainly developed for scRNA-seq data generated by plate-based approaches (e.g., Fluidigm C1). Relative to droplet-based techniques (e.g., 10X Genomics Chromium), plate-based approaches tend to capture fewer cells and more unique transcripts per cell[19]. As such, existing approaches have three key limitations when applied to droplet-based scRNA-seq data. First, existing approaches yield poor point estimates of cell phase due to transcript likelihood distribution choices that do not closely approximate the true generative distribution of droplet-based scRNA-seq data. Second, existing approaches do not quantify the uncertainty of phase estimates. This becomes crucial for interpretation of results from very sparse droplet-based scRNA-seq data. Third, run times of existing approaches scale poorly with the number of cells, making analysis of droplet-based scRNA-seq data untenable for many applications.

To address these unmet needs, we developed Tempo, a Bayesian variational inference approach, for circadian phase inference. Tempo works well for both droplet-based and plate-based scRNA-seq data. Tempo is fast, can incorporate domain knowledge, and yields uncertainty quantifications for the estimated circadian phases. Using both simulated data with ground-truths and real scRNA-seq data, we demonstrate Tempo's ability to achieve state-of-the-art cell phase point estimates and well-calibrated cell phase uncertainty quantifications.

## Results
### Overview of Tempo
Tempo assumes transcript counts of gene $j$ in cell $i$, $X_{ij}$, follow a Negative Binomial distribution. The mean proportion of transcript counts are presumed to follow a 24 h sinusoidal waveform, the gene expression parameters of which are assumed to be shared by all cells. Thus, the mean of transcript counts of gene $j$ in cell $i$ is influenced by two factors: (1) gene-specific parameters, $\beta_j$, describing the shape of the sinusoid and (2) the cell's circadian phase, $\theta_i$. Given the observed data, $\mathbf{X}$, and prior knowledge of all cell and gene parameters, $P(\theta,\beta)$, we seek the posterior distribution of the cell and gene parameters, $P(\theta,\beta|\mathbf{X})$. However, a closed-form solution for $P(\theta,\beta,|\mathbf{X})$ is unknown and estimation using sampling techniques is computationally burdensome. For a computationally efficient solution, Tempo instead proposes an approximate posterior distribution $q(\theta,\beta)$ with differentiable parameters describing its shape. Tempo estimates the true posterior, $P(\theta,\beta|\mathbf{X})$, by maximizing its similarity with the approximate posterior, $q(\theta,\beta)$, through a two-step iterative process (Fig. 1). As input, Tempo requires the observed data, $\mathbf{X}$, prior knowledge $P(\theta,\beta)$, and a list of core clock genes. Tempo uses this information to initialize a list of cycling genes, which only includes the core clock genes at initialization, and the approximate posterior, $q(\theta,\beta)$. The approximate

posterior $q(\theta,\beta)$ is formulated such that only cycling genes contribute information to the approximate posterior estimate of cell phases. In Step 1, Tempo optimizes $q(\theta,\beta)$ to minimize its Kullback-Leibler (KL) divergence with $P(\theta,\beta|\mathbf{X})$ using only information from the current cycling genes. After this step, the marginal of $q(\theta,\beta)$ with respect to $\theta$ can be considered a rough estimate of the cell circadian phase posterior distributions based on only the current cycling genes. In Step 2, Tempo uses the cell phase posterior distributions from Step 1 to identify de novo cyclers. For the set of genes not currently identified as cyclers, approximate gene parameter distributions are fit, conditioned on the cell phase posterior distributions from Step 1. Tempo then selects de novo cycling genes as those best described by phase variation and adds them to the set of current cycling genes. Steps 1 and 2 are repeated until the core clock gene Bayesian evidence worsens or the maximum number of iterations is exceeded. The final result of the algorithm is the optimized approximate joint posterior distribution $q(\theta,\beta)$, which contains information about posterior cell phases and the set of identified de novo cycling genes.

### Evaluations of Tempo on simulated data
We first assessed Tempo's performance on simulated scRNA-seq data generated from Tempo's Negative Binomial count model where sinusoidal gene parameters, including mesor, amplitude, and acrophase, were estimated from a light-dark cycle time course scRNA-seq dataset. This approach was used to simulate scRNA-seq datasets collected from either a single sample of unsynchronized cells or from cells sampled every 4 h over a 24 h light-dark cycle time course (i.e., ZT0, ZT6, ZT12, and ZT18). Details on the estimation of gene parameters and generative model used for simulations can be found in Methods.

Using these simulated scRNA-seq data, we first determined whether Tempo can accurately estimate circadian phase when considering only the core clock genes as input. Tempo was run using a non-informative prior over cell phases. To mimic informative but imperfect gene priors, core clock acrophase prior locations were shifted from their true values. Shifts were drawn from standard normal distributions, scaled by $2 \times \frac{12}{\pi}$ (i.e., standard deviation of 2 h), and added to the true acrophase values to yield acrophase prior locations. The prior clock acrophase scales of the Von Mises distribution were set such that the width of the 95% interval surrounding the prior acrophase locations was 4 h. Cell phase point estimate errors were visualized as empirical cumulative distribution functions (eCDFs). For both unsynchronized and time course datasets, and across a wide range of cell numbers (500–5000 cells) and library sizes (3000–20,000 median unique molecular identifiers (UMIs)), Tempo yields point estimate error eCDFs slightly worse than the optimum (Fig. 2a and Supplementary Figs. 1–9a). The optimum was obtained by computing the maximum likelihood phases using the true generative model as the likelihood model and considering all true cycling genes as input and setting gene parameters to their true values. For comparison, we analyzed the simulated data using existing unsupervised phase inference approaches with run time characteristics suitable for droplet-based scRNA-seq, Cyclops[20] and Cyclum[16]. Cyclops and Cyclum are autoencoder neural network approaches that aim to find a circular projection that maximizes the likelihood of the data. While Cyclops and Cyclum have conceptual similarities, Cyclum uses transformed counts of the individual genes as input, while Cyclops uses principal components of the genes as input. As a baseline, we also included PCA[21], which aims to find orthogonal, linear projections (i.e., principal components) of the data that maximize its likelihood. Using the two principal components that explain the most variation in the data, cell phases can be estimated assuming points in this 2-D space lie along a circle. To assess the effect of feature selection, competing methods were run using two different input gene sets: first, using only the core circadian clock genes, and second, using all genes with pseudobulk UMI proportions, i.e., relative abundances, greater than $10^{-5}$. These

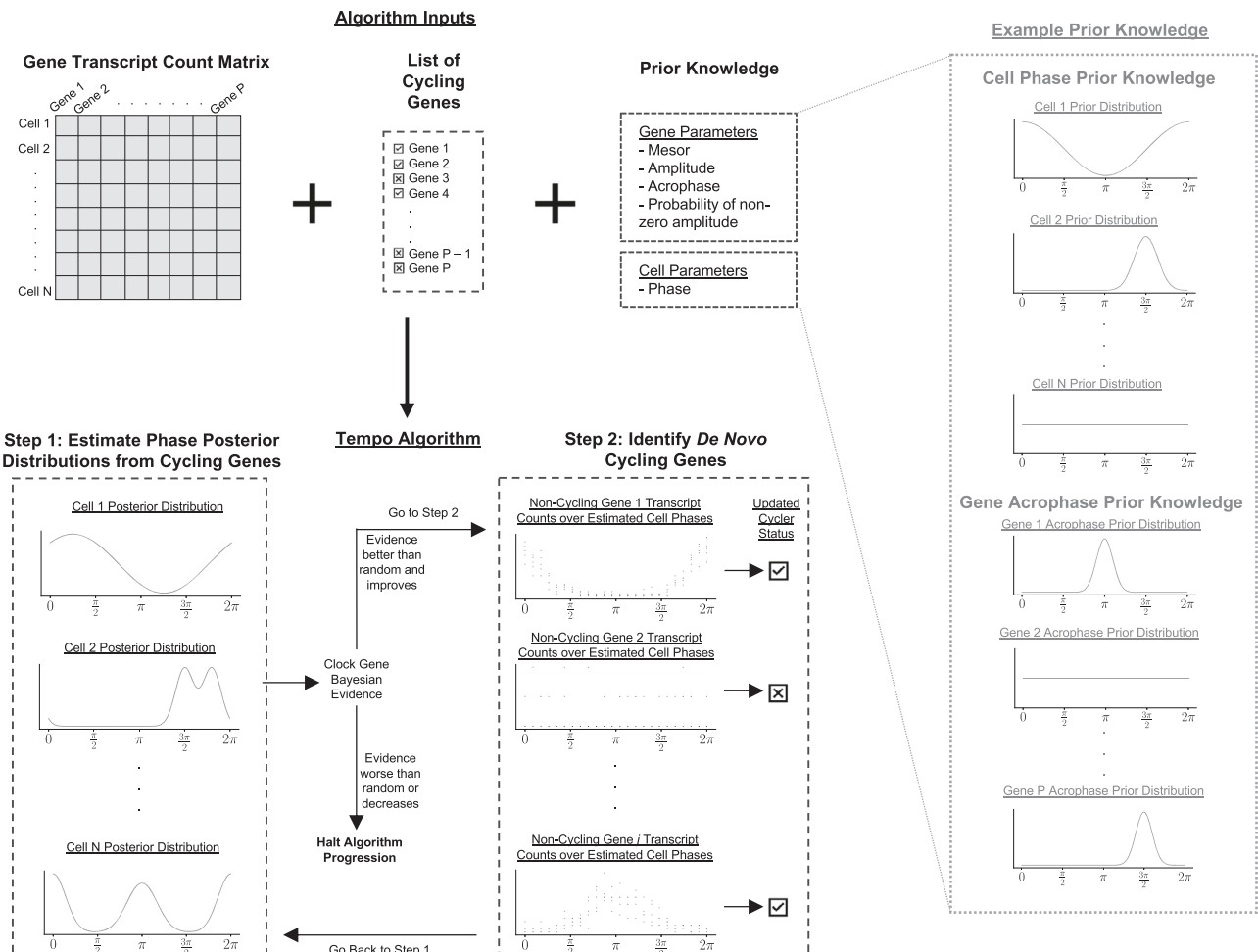

**Fig. 1 | Tempo model overview.** As input, users supply a cell transcript count matrix, list of cycling genes (e.g., circadian clock genes), and prior knowledge about the cell and gene parameters. Using the user-specified cycling genes, the count data and prior knowledge, in Step 1 Tempo computes approximate posterior distributions of the cell circadian phases. Using these, in Step 2 Tempo identifies de novo cycling genes with transcript counts that are well-explained by circadian phase variation. Tempo repeats Steps 1 and 2 until either the Bayesian evidence of the user-supplied cycling genes (e.g., circadian clock genes) worsens relative to previous iterations of the algorithm or is worse than random. Source data are provided as a Source Data file.

competing approaches generally yielded non-random performance, but demonstrated a stark decrease in performance for data with smaller library sizes. As additional baselines, we also compared different methods to two naïve approaches: (1) drawing cell phases from a circular uniform distribution, and (2) drawing cell phases from a single point.

We further probed whether the uncertainty quantifications associated with Tempo's phase estimates using the core clock genes alone were well-calibrated. We assessed the relationship between the confidence of the approximate posterior's credible interval and the corresponding fraction of intervals containing the true cell phase. Credible intervals were computed using the Highest Density Region approach[22]. Encouragingly, this analysis (Fig. 2b and Supplementary Figs. 1–9b) suggests Tempo's uncertainty quantifications are well-calibrated for both unsynchronized and time course data. Tempo's credible intervals are slightly conservative, which reflects the propagation of uncertainty from the gene parameters.

Given that Tempo can estimate cell phase from the core clock genes alone, we next assessed the feasibility of de novo cycler detection and the potential use of de novo cyclers to improve cell phase point estimates. Tempo was run on the simulated datasets considering all genes as input so that de novo cyclers could be detected and included into cell phase estimates. For comparison, Cyclops, Cyclum, and PCA were run considering all genes as input. For both

unsynchronized and time course datasets, and across a range of simulation settings, Tempo identifies de novo cycling genes with high specificity and sensitivity (Fig. 2c and Supplementary Figs. 1–9c). Notably, incorporating de novo cyclers with core clock genes improves cell phase point estimates (Fig. 2d and Supplementary Figs. 1–9d). In comparison, competing methods did not see a significant improvement in point estimates when considering all genes as input. Phase uncertainties remained well-calibrated, albeit more conservative, when incorporating de novo cyclers (Fig. 2e and Supplementary Figs. 1–9e). This suggests that de novo cycler detection can be a valuable tool for circadian phase estimation.

We further assessed the stability of Tempo's predictions. Tempo's results are stochastic due to the sampling required to compute the objective function for both cell phase estimation and de novo cycler detection. It is crucial to validate that the default number of samples used to compute the objective function yields stable results. To assess method stability, methods were run on the same simulated dataset multiple times. For each method, the circular standard deviation (reported in hours) of all cells was computed and visualized as a distribution. For comparison, circular standard deviation distributions were also computed by randomly drawing cell phases from a circular uniform distribution. Tempo's median circular standard deviation was less than 1 h for all simulation settings for which stability was evaluated (Fig. 2f and Supplementary Figs. 1f, 2f and 7f). Of note, Cyclum and

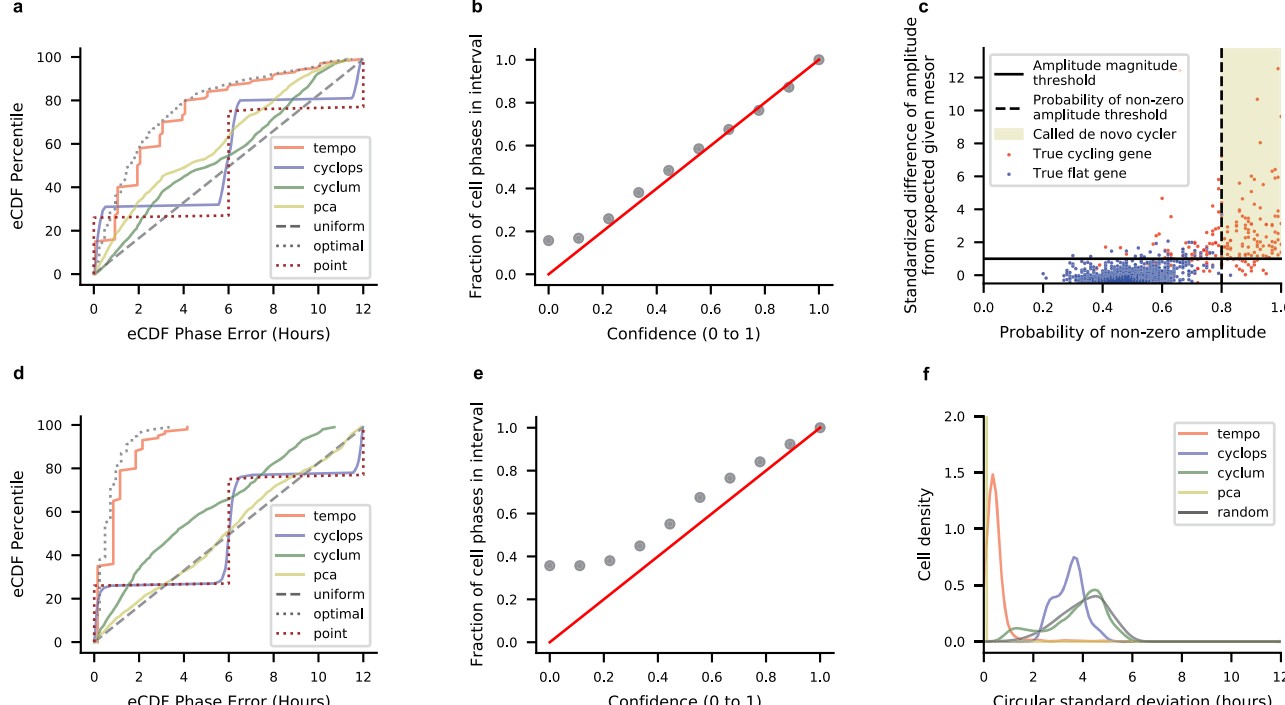

**Fig. 2 | Results on a simulated scRNA-seq dataset of 1000 cells collected at CT0, CT6, CT12, and CT18 with mean library size of 10,000 UMI. a** Empirical cumulative distribution function (eCDF) of the errors for each method's cell phase point estimates, where all methods were run using the true core clock genes as input. **b** Calibration of Tempo's uncertainty estimates when run using the true core clock genes as input. **c** Tempo's de novo cycler detection procedure. The x-axis represents the maximum a posteriori (MAP) fraction of samples with non-zero amplitude for a given gene, and captures whether a gene is better described by sinusoidal or flat variation over the circadian cycle. The y-axis statistic measures deviation of a gene's MAP amplitude from its expected MAP amplitude given its MAP mesor, reported in terms of a Pearson residual. Large positive values indicate a gene has a larger amplitude than expected given its mesor. Details of the Pearson residual computation can be viewed in Supplementary Methods 8. **d** eCDF of the errors for method cell phase point estimates, where methods were run considering all genes as input. **e** Calibration of Tempo's uncertainty estimates when run considering all genes as input. **f** Method stability analysis. Methods were run five times (considering all genes as input) on the dataset. The circular standard deviation of predictions for each cell was computed and visualized as a distribution. Source data are provided as a Source Data file.

Cyclops yielded highly unstable results for the simulation settings evaluated.

Lastly, we evaluated how Tempo's assumption of pure 24 h component sinusoidal waveforms affects its performance on simulated data with more realistic waveforms. To assess this, simulated data were generated using waveforms from a bulk aorta circadian time course RNA microarray dataset sampled every 2 h over 48 h generated by Zhang et al.[23]. JTKCycle[24] was run on these data with a fixed period of 24 h, and genes with Benjamini-Hochberg $q$ values <0.05 were considered true positive cyclers. In total, 4800 cells were simulated with a median library size of 5000 UMIs. As a general measure of each gene's temporal signal, we computed the likelihood ratio of the true waveform over a flat waveform (Fig. 3a). Moreover, for each true cycling gene we computed the strength of the 24 h sinusoidal component relative to other sinusoidal components. We refer to this metric as the circadian FFT fraction. Further details on the simulations and analysis can be viewed in Methods. Our analyses suggest the waveforms of the core circadian clock genes are among the most similar to pure 24 h sinusoids (Fig. 3b). Tempo's phase point estimates made based on the core clock genes are largely unaffected when using their true waveforms, as errors closely mirror that of the theoretical optimum (Fig. 3c). Moreover, cell phase uncertainties remain well-calibrated (Fig. 3d). Running Tempo with de novo cycler detection, Tempo called 25 de novo cycling genes, all of which were true cyclers. The called de novo cyclers were among the true cycling genes with the most temporal signal (Fig. 3e). The waveforms of called de novo cyclers had modestly stronger 24 h sinusoidal components than those of all true cyclers (Fig. 3f). However, the waveforms of called cyclers had notably less pure 24 h sinusoidal components than those of the core circadian clock genes *Dbp*, *Nr1d1*, and *Arntl*. Undetected cycling genes with similar temporal strength to that of the detected cycling genes demonstrate similarly pure 24 h sinusoidal components (Supplementary Fig. 3g). This suggests that Tempo's 24 h sinusoidal component assumption likely does not strongly affect de novo cycler detection sensitivity. On these data, incorporation of de novo cyclers improves Tempo's cell phase point estimates (Fig. 3h). For example, 62% of estimates lie within 3 h of the true cell phase based on the clock alone; this improves to 72% when incorporating de novo cyclers. However, unlike the results on the data simulated with pure 24 h component sinusoidal waveforms, results on these data suggest Tempo's point estimates incorporating de novo cyclers are suboptimal. Incorporation of de novo cyclers yields uncertainty estimates that remain well-calibrated (Fig. 3i). Altogether, these results suggest Tempo's assumption of 24 h component sinusoidal waveforms is reasonable for estimating cell phase from the core circadian clock genes and for de novo cycler detection. However, this assumption is suboptimal for using de novo cyclers to improve phase estimates.

## Tempo accurately estimates circadian phase from real scRNA-seq data

While encouraged by Tempo's performance on the simulated data, Tempo's likelihood distribution exactly matches the generative distribution of the simulated data. In this sense, Tempo's superior performance relative to other methods is, perhaps, unsurprising. We next assessed whether the quality of Tempo's circadian phase estimates generalized to real droplet-based scRNA-seq data. We generated a

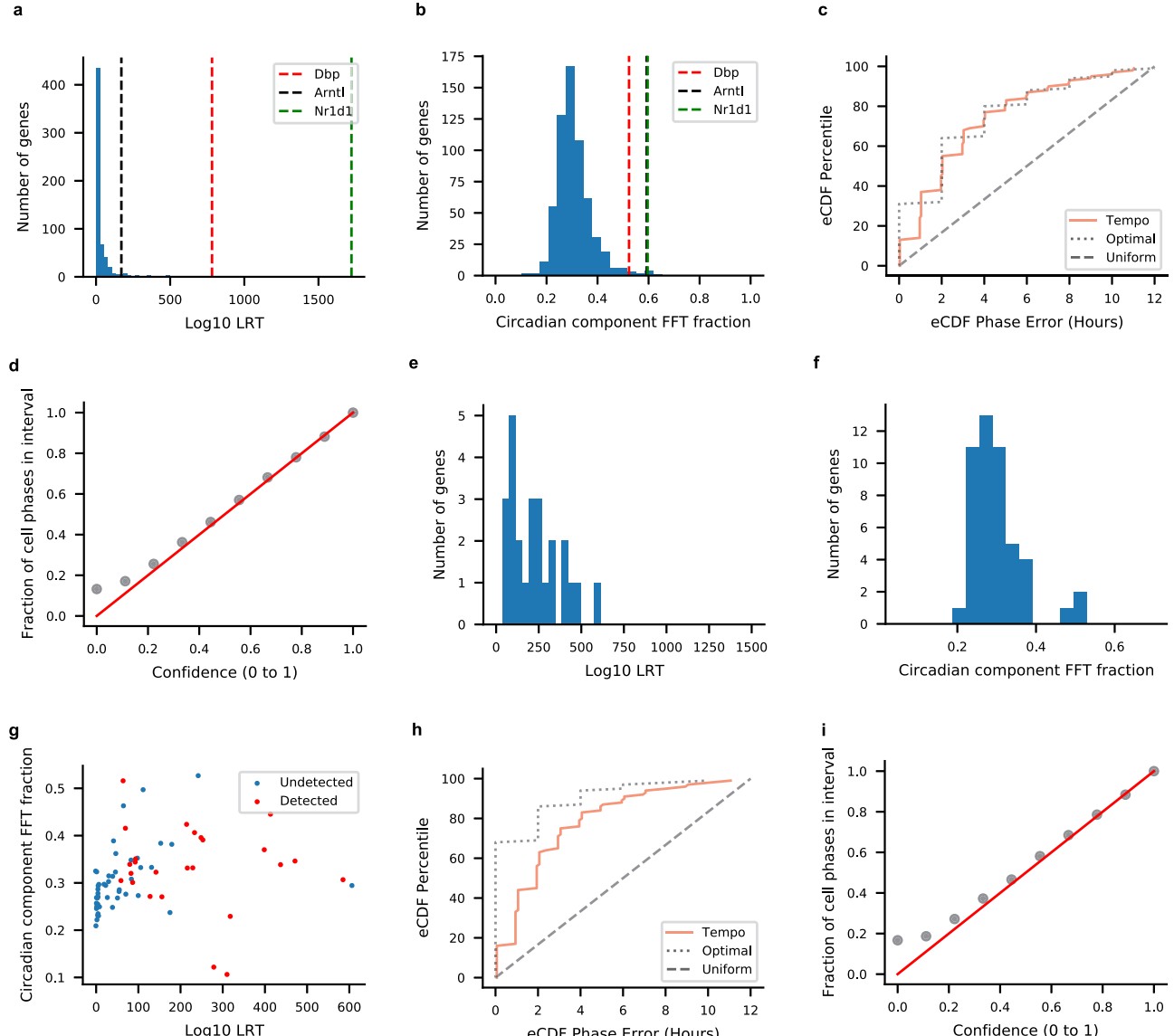

**Fig. 3 | Tempo results on simulated scRNA-seq with realistic waveforms.**
**a** Distribution of true cycler temporal signal strength, measured according to the likelihood ratio test statistic (LRT) of the true waveform vs. a flat waveform. **b** Distribution of true cycler 24 h sinusoid component strength, measured according to the circadian fast Fourier transform (FFT) fraction. **c** eCDFs of the error of Tempo's cell phase point estimates and **d** uncertainty calibration when run with the core clock genes alone. **e** Distribution of de novo cycler temporal signal strength. **f** Distribution of the strength of the 24 h sinusoidal component for genes called as de novo cyclers by Tempo. **g** Bivariate distributions of cycler temporal strength and 24 h sinusoidal component strength for true cyclers detected or undetected as de novo cyclers. **h** eCDFs of the error of Tempo's cell phase point estimates and **i** uncertainty calibration when run with the core clock genes and de novo cyclers. Source data are provided as a Source Data file.

deeply sequenced scRNA-seq dataset using the 10X Genomics Chromium platform from mouse aorta collected every 4 h (i.e., ZT0, ZT6, ZT12, and ZT18) over a 24 h light-dark cycle. This high-quality dataset yielded 18,863 vascular SMCs, 3135 fibroblasts, 288 endothelial cells, and 287 macrophages with median library sizes of 13,646, 7412, 6846.5, and 7389 UMIs, respectively. To benchmark Tempo on this dataset, we compared its performance with Cyclops, Cyclum, and PCA. To assess the effect of feature selection, competing methods were run using two different input gene sets: first, using only the core circadian clock genes, and second, using all genes (with pseudobulk UMI proportions greater than $10^{-5}$) to evaluate if including de novo cyclers helps cell phase estimates. Tempo was run using a non-informative prior over cell phases. Additional details on the initialization of the core clock gene priors are provided in Methods. The resulting cell phase predictions for each individual time point can be viewed in Fig. 4 and Supplementary Figs. 10–13.

As a diagnostic tool, Tempo measures the Bayesian evidence of core clock expression associated with the estimated parameters. Tempo compares this relative to the evidence associated with random parameters estimated on a permuted core clock count matrix, and summarizes improvement over random via their ratio, also referred to as the Bayes factor. Further details of this procedure can be viewed in Supplementary Methods 6 and 7. Bayes factors greater than 1 indicate improvement over random. This diagnostic tool suggests Tempo's predictions on these data were highly non-random. The Bayes factors were $10^{17,896}$, $10^{2257}$, $10^{110}$, and $10^{184}$ for the SMCs, fibroblasts, endothelial cells, and macrophages, respectively.

For each cell type, we first compared the circadian phase point estimates of individual cells to their sample collection phase in the light-dark cycle. The distribution of the difference between the two phases over all cells was visualized as an eCDF (Fig. 5a and Supplementary Figs. 14–20a). For all cell types and input gene sets, Tempo's

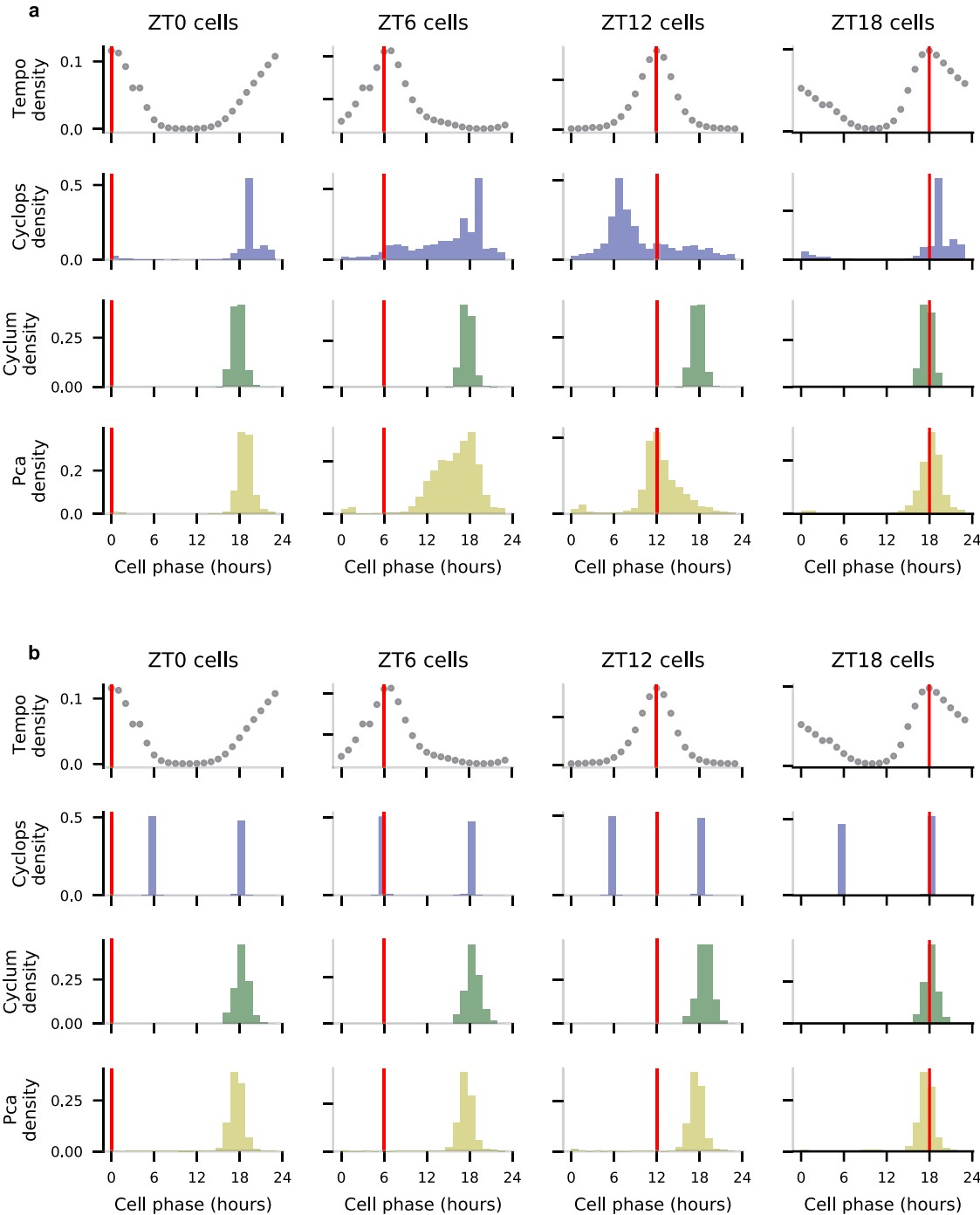

**Fig. 4 | Density of method cell phase predictions for aorta SMCs at various sample collection times, reported in terms of Zeitgeber time (ZT).** Tempo's densities represent the pseudobulk approximate posterior distributions at each sample collection time point. Competing method densities represent method point estimates. Vertical red lines represent the anticipated cell phase given the sample collection time. **a** Method cell phase predictions densities when run using only the core clock genes. **b** Method cell phase predictions densities when run using all genes as input. Source data are provided as a Source Data file.

point estimates demonstrate a substantial improvement over the alternative approaches we analyzed (Table 1). Moreover, on these data Tempo demonstrates well-calibrated uncertainties (Fig. 5b and Supplementary Figs. 14–20b), suggesting its uncertainty quantification is meaningful and can aid interpretation of results.

Phase inference methods were also applied to time course droplet-based scRNA-seq data of 18,378 mouse hepatocytes from Droin et al.[6]. Tempo's predictions on these data were also better than random, as the ratio of Tempo's core clock expression Bayesian evidence over random was $10^{1761}$. On these sparser data (median library

size of 1965 UMIs), Tempo again demonstrates improved point estimates over competing methods and well-calibrated uncertainties (Supplementary Figs. 21 and 22a, b).

The above evaluations assume each cell's circadian phase is equal to its sample collection phase in the light-dark cycle. While we anticipate the true cell phase is close to the sample collection phase in young, healthy mice, the collection phase may not exactly equal the true circadian phase of individual cells if they are imperfectly synchronized. As such, the point estimates were evaluated on light-dark cycle data according to two additional criteria independent of single-

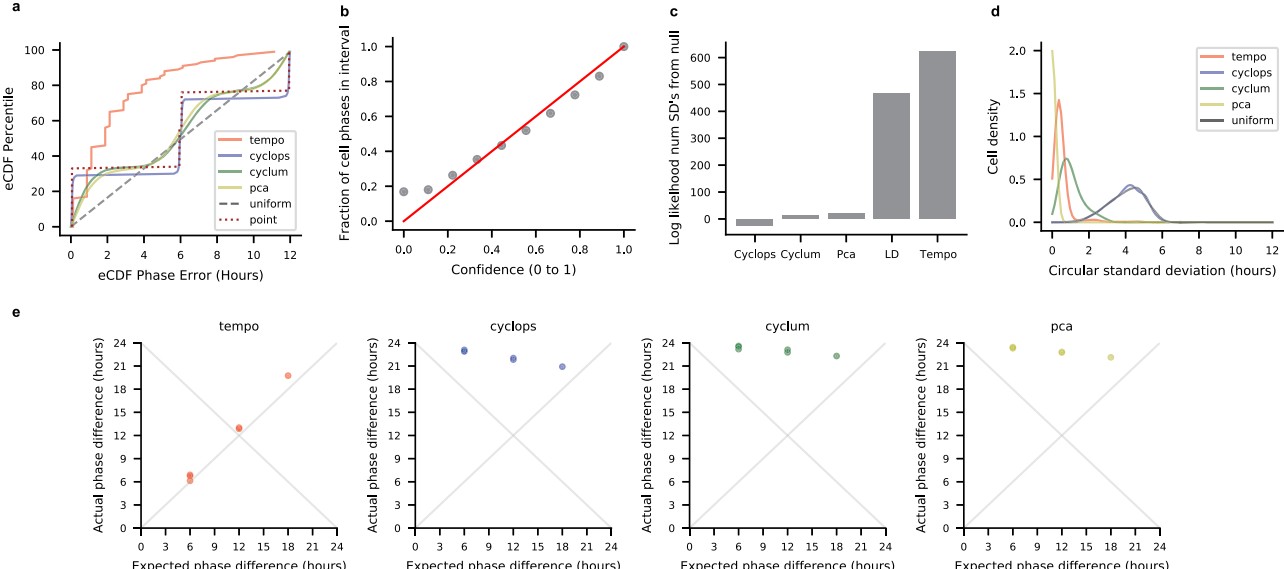

**Fig. 5 | Method results (considering all genes as input) on light-dark cycle aorta smooth muscle cells.** Treating the sample collection phase in the light-dark cycle as the true cell circadian phases: **a** eCDF of the errors for each method's cell phase point estimates, **b** Calibration of Tempo's uncertainty estimates. **c** Method out of sample core clock gene likelihood analysis. LD corresponds to treating sample collection times as the true cell phases. Out of sample core clock likelihoods were computed for each method, and reported in terms of standard deviations from the median of a distribution of likelihoods associated with phase assignments drawn from a random uniform distribution. **d** Method stability analysis. Each methods was

run five times on the dataset. The circular standard deviation of predictions for each cell was computed and visualized as a distribution. **e** Method relative shift analysis. Each dot represents a pair of sample collection times in the light-dark cycle (e.g., all six possible pairs of ZT0, ZT6, ZT12, ZT18), and conveys the relationship between the expected phase difference between a pair of time points and the actual phase difference for each method. As the phase difference is a circular random variable, methods with points lying along either $y = x$ or $y = 24 - x$ denote perfect performance. Source data are provided as a Source Data file.

cell phase ground truths. First, the difference between cell phase estimate distributions for any pair of sample collection phases should equal the difference in the collection phases. Based on this rationale, we calculated the expected relative shift in phase distributions across pairs of sample collection times and their respective cells. As shown in Fig. 5e and Supplementary Figs. 14–16e, 17–22d, Tempo recapitulated the expected relative shift for all cell types. Alternative approaches often did not capture the anticipated relative phase shift between sample collection times, regardless of whether they were run with the core clock genes or the full gene set. Second, the method with phase estimates closest to the truth should best explain core clock expression, in terms of likelihood. Moreover, phase estimate parameters should generalize to explaining the expression of clock gene transcripts in unseen cells. As such, each method was evaluated for their ability to explain core clock expression on a holdout set of cells, as measured by the likelihood of core clock expression using the method's cell phase estimates (Fig. 5c and Supplementary Figs. 14–16c). Details of the computation of holdout cell core clock expression likelihood can be viewed in Methods. Tempo explained holdout clock expression the best across all cell types evaluated. Intriguingly, for the aorta SMCs and fibroblasts, Tempo explains core clock expression better than sample collection phase. This suggests Tempo may meaningfully identify out-of-phase cells collected over light-dark cycles.

The stability of point estimates was assessed by running each method five times on each dataset. For each method, the circular standard deviation (reported in hours) of all cells was computed and visualized as a distribution (Fig. 5d and Supplementary Figs. 14–16d, 17–22c). For comparison, circular standard deviation distributions were also computed by randomly drawing cell phases from a circular uniform distribution. Tempo's median circular standard deviation was less than 1 h for all datasets, with exception of the aorta endothelial cells. For the endothelial cells, 27% of cells had circular standard deviations <1 h (Supplementary Fig. 23a). Cells with higher posterior

certainty had more stable estimates (Supplementary Fig. 23b) than cells with less certainty (Supplementary Fig. 23c). Relative to the SMCs and fibroblasts, the endothelial cells had many fewer cells (18,863, 3135 fibroblasts, and 288 cells for SMCs, fibroblasts, and endothelial cells, respectively). As such, Tempo's relative instability on the endothelial cells could be partially explained by the small cell counts. Nonetheless, it was unexpected that the endothelial cell predictions were more unstable than that of the macrophages, as the data for the two cell types had similar technical characteristics; the data for the two cell types had similar cell counts (288 and 287 cells for endothelial cells and macrophages, respectively) and median library sizes (6846.5 and 7389 UMI for endothelial cells and macrophages, respectively). Further inspection suggested that stability differences between the two cell types may be explained by weaker core circadian clock expression in endothelial cells. Core clock gene transcripts were detected in a smaller fraction of endothelial cells and had smaller pseudobulk means, suggesting smaller mesors in endothelial cells (Supplementary Fig. 23d, e). Moreover, clock gene transcripts exhibited smaller standard deviations (Supplementary Fig. 23f), suggesting smaller amplitudes in endothelial cells. In general, Cyclop's and Cyclum's estimates were unstable, and their circular standard deviation distributions heavily overlapped with the circular uniform distribution for many of the cell types and sets of input genes. However, Cyclum's stability was notably dataset dependent, as it showed good stability on the aorta SMCs. As such, Cyclum's stability may heavily depend on the dataset characteristics and choice of hyperparameters.

## Tempo identifies de novo cycling genes from real scRNA-seq data

De novo cycling genes were called by Tempo for all real datasets. For the real circadian light-dark cycle datasets 189, 109, 87, 28, and 117 de novo cyclers were called for the aorta SMCs, aorta fibroblasts, aorta endothelial cells, aorta macrophages, and liver hepatocytes, respectively. The quality of these cyclers was assessed by two criteria. First,

**Table 1 | Comparison of method median phase error in hours**

| Method | Gene set | Aorta SMCs | Aorta fibroblasts | Aorta endothelial cells | Aorta macrophages | Liver hepatocytes |
|--------|----------|------------|-------------------|-------------------------|-------------------|-------------------|
| Tempo | All | 1.88 (0.03) | 2.53 (0.36) | 3.39 (0.4) | 2.99 (0.01) | 3.78 (0.39) |
| Cyclops | All | 5.73 (0.29) | 5.84 (0.08) | 5.65 (0.54) | 5.35 (0.24) | 4.66 (0.3) |
| Cyclops | Core clock | 4.62 (1.22) | 4.17 (0.74) | 5.72 (0.03) | 4.08 (0.14) | 4.26 (0.27) |
| Cyclum | All | 5.47 (0.44) | 5.79 (0.13) | 5.74 (0.23) | 5.69 (0.31) | 5.15 (0.53) |
| Cyclum | Core clock | 5.52 (0.35) | 4.79 (0.69) | 5.0 (0.45) | 4.18 (0.31) | 5.65 (0.42) |
| PCA | All | 5.69 (0.0) | 6.0 (0.0) | 5.84 (0.0) | 5.64 (0.0) | 5.75 (0.0) |
| PCA | Core clock | 3.39 (0.0) | 3.99 (0.0) | 5.58 (0.0) | 4.41 (0.0) | 5.91 (0.0) |

Entries denote the mean of median phase errors across five independent runs for each method (in hours). Parenthetical values denote the standard deviation of median phase errors across the five runs (in hours). Of note, PCA is deterministic and its predictions do not differ between runs.

for each cell type, we ran Step 2 of Tempo assuming the cell phases were equal to their sample collection phase and called de novo cyclers. When run in a fully unsupervised manner, Tempo's de novo cyclers should be enriched for cyclers that were called when cell phases were fixed to their sample collection phase. Indeed, de novo cyclers were enriched for these cyclers for all cell types evaluated (Fig. 6a–e). Second, we expect that de novo cyclers detected from cell types are enriched for cycling genes detected from bulk datasets from the same tissue. Using the bulk aorta and liver datasets from Zhang et al.[23], genes with JTKCycle $q$ values <0.05 were considered to be true bulk cyclers. Strong enrichment was observed for the aortic SMCs, aortic fibroblasts, and liver hepatocytes (Fig. 6f–j). While more modest enrichment was observed for aortic macrophages (empirical $p$ value = 0.1165) and no enrichment for aortic endothelial cells (empirical $p$ value = 0.6759), these cell types compose a smaller proportion of the tissue than SMCs and fibroblasts[25]. As such, we would expect less concordance with the bulk aorta results for these cell types. Altogether, these results suggest Tempo reliably identifies de novo cycling genes in real circadian scRNA-seq data.

In contrast to its performance on the simulated data with only 24 h sinusoidal components, Tempo favored cell phase estimates based on the core clock genes alone rather than those including de novo cyclers for all real light-dark cycle cell type datasets. While Tempo's 24 h component sinusoidal assumption is adequate for identifying de novo cyclers, these results suggest this assumption is limiting for incorporating de novo cyclers to improve phase estimates.

## Discussion

Single-cell transcriptomics offers an unprecedented opportunity to improve our understanding of circadian transcription. Nonetheless, its impact has been limited by the assumption that sample collection times reflect cell circadian phases. In lieu of widespread experimental approaches that jointly measure single-cell phase and transcriptomes, computational phase inference tools can be applied to estimate cell phases from scRNA-seq directly. However, existing tools yield poor phase estimates and do not quantify estimation uncertainty.

To address these challenges, we developed Tempo, a Bayesian algorithm for single-cell phase inference from scRNA-seq data. Through a combination of both simulated and real data analyses, we demonstrate that Tempo yields state-of-the-art point estimates. Moreover, Tempo empowers better phase estimate interpretation through well-calibrated uncertainty quantifications and measurement of improvement over random phase assignments. While developed for circadian phase inference, Tempo's framework likely generalizes to other cyclical processes, such as the cell cycle. Lastly, Tempo's run time characteristics are amenable to larger droplet-based scRNA-seq datasets. Across all datasets analyzed, Tempo completed within 1 h using a desktop CPU with a 3.2 GHz Intel Xeon W processor and 32GB RAM.

Given Tempo's performance, we are encouraged by its immediate potential for use in circadian research. More specifically, our tool may make it possible to characterize circadian transcription parameters using single samples of unsynchronized cell cultures. This experimental paradigm can enable cost-effective circadian transcription studies of human subjects and be used to study the role of cell-cell interactions in circadian transcription. Further, our tool can help researchers study circadian phase heterogeneity and its biological determinants in tissue contexts (e.g., spatial location) from time course data. Indeed, our analyses suggest Tempo meaningfully identifies out-of-phase cells in mouse aorta (Fig. 5c and Supplementary Figs. 14–16c). Lastly, cell populations demonstrate cell-cell heterogeneity in circadian gene expression parameters, such as acrophase and amplitude of clock genes etc. This Bayesian approach naturally captures this biological variation and facilitate its interpretation.

Although Tempo has achieved several advances in unsupervised phase inference, opportunities for future improvements exist. First, de novo cyclers do not improve point estimates in real scRNA-seq datasets. While incorporating de novo cyclers improves point estimates in simulated data, de novo cyclers decreased evidence of core clock expression in the real scRNA-seq datasets we analyzed. Our simulated analyses suggest this may be due, in part, to the assumption that the expression means of CCGs follow 24 h sinusoidal patterns across the circadian cycle. Future efforts might rely on approaches that can model more flexible CCG waveforms. Second, our method does not explicitly model the contribution of technical effects to expression variation. While less important for applications to single-sample unsynchronized scRNA-seq data, this becomes more necessary for data collected as multiple samples over time. Lastly, Tempo assumes the sinusoidal parameters are shared by all input cells. Nonetheless, parameters (e.g., gene amplitudes) may vary across subpopulations of the input cells. As a Bayesian method, Tempo naturally handles such situations by modeling additional variance in the gene and cell parameter estimates. However, a more ideal solution may be to use a function to map continuous measures of cell state (i.e., a low-dimensional embedding) to the sinusoidal parameters.

While we await widespread experimental approaches to pair single-cell clock reporters with transcriptomics, it is important to keep in mind that clock reporters with even zero measurement error will contain phase uncertainty due to the inherent stochasticity of the clock. Thus, single-cell reporter phases may be best used as prior knowledge to unsupervised phase inference algorithms, such as Tempo.

In summary, we developed Tempo, a Bayesian algorithm for circadian phase inference from scRNA-seq data. Tempo yields state-of-the-art point estimates of circadian phase, and well-calibrated uncertainties. Well-calibrated uncertainties will enable investigators to understand the robustness of cell phase estimates made from highly sparse scRNA-seq data and to account for uncertainty in downstream analyses. Further, the quality of Tempo's phase estimates may make it

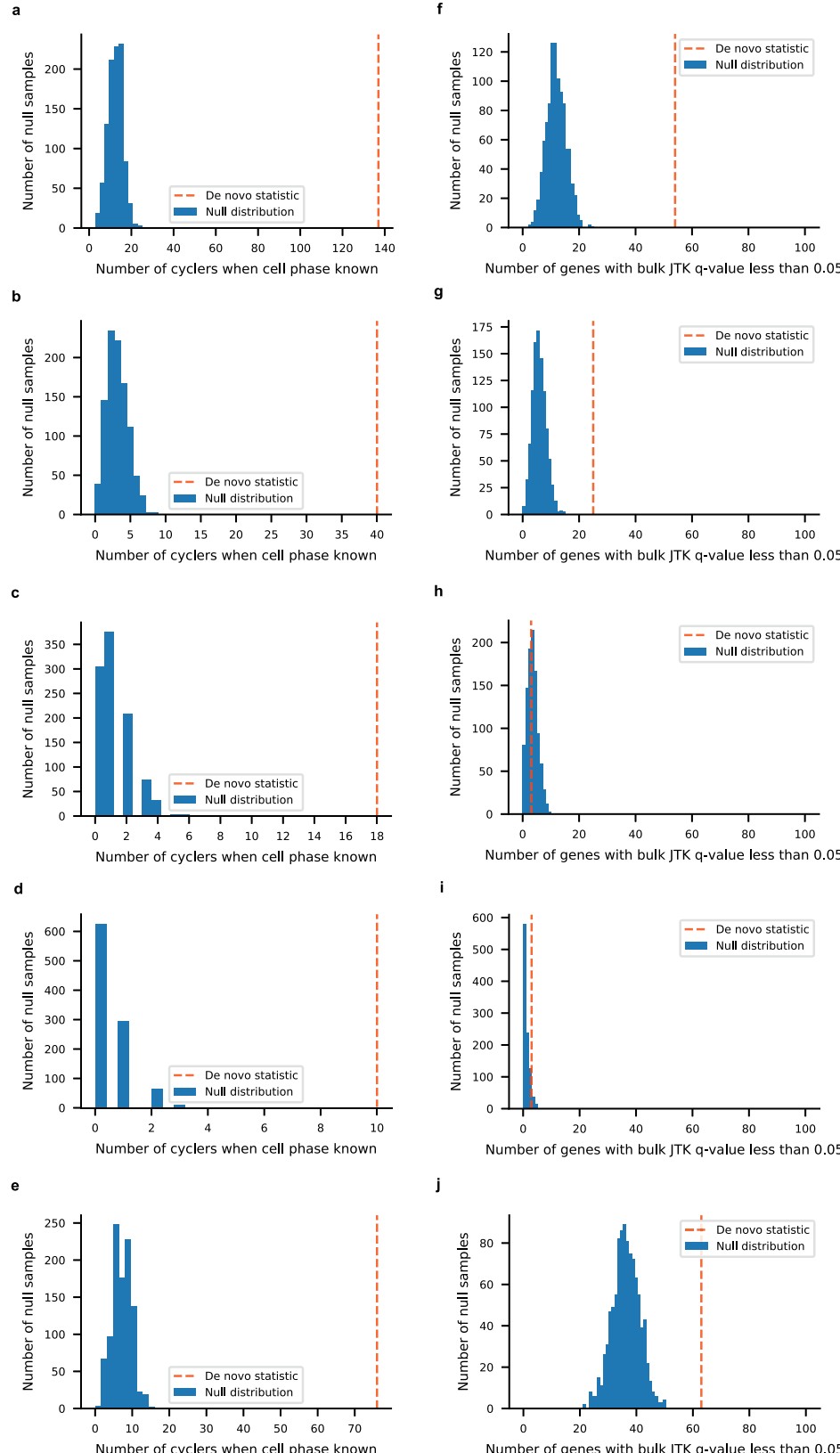

**Fig. 6 | Tempo called de novo cycler enrichment.** Enrichment in called cyclers when cell phase set to sample collection times for **a** aorta SMCs, **b** aorta fibroblasts, **c** aorta endothelial cells, **d** aorta macrophages and **e** liver hepatocytes. Enrichment in bulk cyclers (JTKCycle *q* value <0.05) for **f** aorta SMCs, **g** aorta fibroblasts, **h** aorta endothelial cells, **i** aorta macrophages and **j** liver hepatocytes. Source data are provided as a Source Data file.

possible to identify out-of-phase cells from tissue collected over time courses and to estimate circadian parameters from unsynchronized cell populations using scRNA-seq.

## Method

### Ethical compliance

All research described in the manuscript adheres to relevant ethical regulations. For the aorta data generated, Animal Care Protocol #805035 was approved by the Institutional Animal Care and Use Committee of the University of Pennsylvania.

### Tempo model

**Likelihood model.** As input, Tempo requires an $n \times p$ transcript count matrix $\mathbf{X}$, where $n$ is the number of cells and $p$ is the number of genes. For gene $j$ in cell $i$, the UMI count $X_{ij}$ is assumed to follow a Negative Binomial (NB) distribution. The expected log proportion, $\log \lambda_{ij}$, of gene $j$'s transcripts in cell $i$ is defined by a sinusoid with four parameters: (1) the mesor, $\mu_j$, which controls the mean of a gene's proportion over the circadian cycle, (2) the amplitude, $A_j$, or the maximum deviation of the gene's proportion from the mesor over the circadian cycle, (3) the acrophase, $\phi_j$, or the peak time of the gene's proportion over the circadian cycle, and (4) an indicator, $Q_j$, describing whether the gene has non-zero amplitude. These sinusoidal gene parameters are assumed to be shared across all cells. As such, observed expression differences across cells are explained by differences in (1) latent cell phase, $\theta_i$, (2) cell library size, $L_i$, and (3) random variation described by the Negative Binomial distribution.

The distribution of $X_{ij}$ is modeled as:

$$X_{ij} \sim \mathrm{NB}\left(L_i \lambda_{ij}, \delta_{ij}\right), \tag{1}$$

where:

$$\mathrm{E}\left[X_{ij}\right] = L_i \lambda_{ij}, \tag{2}$$

$$\mathrm{Var}\left(X_{ij}\right) = \mathrm{E}\left[X_{ij}\right] + \delta_{ij} \times \left(\mathrm{E}\left[X_{ij}\right]\right)^2 \tag{3}$$

and:

$$\log\left(\lambda_{ij}\right) = \mu_j + Q_j A_j \cos\left(\theta_i - \phi_j\right), \tag{4}$$

$$\delta_{ij} = \mathrm{g}_{\boldsymbol{\zeta}}\left(\lambda_{ij}\right), \tag{5}$$

where $\mathrm{g}_{\boldsymbol{\zeta}}(\lambda_{ij})$ is a deterministic polynomial function parameterized by $\boldsymbol{\zeta}$ (shared by all cells and genes) describing the relationship between transcript proportion $\lambda_{ij}$ and the dispersion $\delta_{ij}$. Details on the estimation of $\boldsymbol{\zeta}$ can be found in Supplementary Methods 1.

**Prior knowledge of gene and cell parameters.** Prior knowledge may be known about cell phases (e.g., based on single-cell clock gene reporters or cell collection time); in this case, users can specify prior knowledge about the phase of cell $i$ as a Von Mises distribution (a circular analog to the normal distribution):

$$\mathrm{P}(\theta_i) = \mathrm{Von\ Mises}\left(\theta_i^{(\mathrm{loc})}, \theta_i^{(\mathrm{scale})}\right) \tag{6}$$

In the absence of prior knowledge about cell phases, Tempo uses a non-informative Hyperspherical Uniform[26] prior for each cell phase by default.

For the gene parameters, prior knowledge about the mesor of gene $j$ can be specified as a normal distribution:

$$\mathrm{P}(\mu_j) = \mathrm{Normal}\left(\mu_j^{(\mathrm{loc})}, \mu_j^{(\mathrm{scale})}\right) \tag{7}$$

In practice, we use an empirical Bayesian approach to set $\mu_j^{(\mathrm{loc})}$ equal to the log proportion of transcripts for each gene.

Prior knowledge about the acrophase of gene $j$ may exist, (e.g., from bulk circadian transcriptomics data), in which case prior knowledge can be specified in terms of a Von Mises distribution. In the case of the core circadian clock genes, prior knowledge is typically known. Otherwise, by default Tempo assumes a non-informative Hyperspherical Uniform prior.

The algorithm additionally requires the user to specify a reference gene, the peak time of which defines the start of the circadian cycle. To enforce this, the prior acrophase distribution of the reference gene is set to a point mass centered at 0 radians. By default, the algorithm uses the core circadian clock gene, *Arntl*, as the reference gene defining the start of the cycle.

Prior knowledge about the amplitude of gene $j$ is specified as a transformed Beta distribution:

$$\mathrm{P}(A_j) = \mathrm{Beta}\left(A_j^{(\alpha)}, A_j^{(\beta)}\right) \times \left(A^{(\mathrm{max})} - A^{(\mathrm{min})}\right) + A^{(\mathrm{min})} \tag{8}$$

where $A^{(\mathrm{min})}$ and $A^{(\mathrm{max})}$ denote the minimum and maximum possible amplitude (which is shared by all genes). By default, Tempo sets $A_j^{(\alpha)} = A_j^{(\beta)} = 1$, which assumes a non-informative prior over the domain of possible amplitude values.

Prior knowledge about whether a gene $j$ has non-zero amplitude is specified in terms of a hierarchical Beta-Bernoulli:

$$\mathrm{P}(\gamma_j) = \mathrm{Beta}\left(\gamma_j^{(\alpha)}, \gamma_j^{(\beta)}\right) \tag{9}$$

$$\mathrm{P}(Q_j | \gamma_j) = \mathrm{Bernoulli}\left(\gamma_j\right) \tag{10}$$

where samples of $\gamma_j$ denote the success probability of a gene having non-zero amplitude and the user specifies the shape parameters of the Beta distribution. For genes not part of the user-specified list of core clock genes, Tempo sets $\gamma_j^{(\alpha)} = \gamma_j^{(\beta)} = 1$, by default.

**Approximate posterior inference.** Using our prior knowledge of the cell and gene parameters and the observed data, we seek the following joint posterior distribution of our cell and gene parameters:

$$\mathrm{P}(\boldsymbol{\theta}, \boldsymbol{\beta} | \mathbf{X}) \propto \mathrm{P}(\mathbf{X} | \boldsymbol{\theta}, \boldsymbol{\beta}) \mathrm{P}(\boldsymbol{\theta}, \boldsymbol{\beta}) = \mathrm{P}(\mathbf{X} | \boldsymbol{\theta}, \boldsymbol{\beta}) \mathrm{P}(\boldsymbol{\theta}) \mathrm{P}(\boldsymbol{\beta}) \tag{11}$$

where $\boldsymbol{\theta}$ is an $n$-dimensional vector containing the phase for each cell, $\boldsymbol{\beta}$ is a $p$-dimensional vector containing the parameters for each gene (i.e., $\beta_j = (\mu_j, A_j, \phi_j, Q_j, \gamma_j)$), and:

$$\mathrm{P}(\boldsymbol{\beta}) = \prod_{j=1}^{p} \mathrm{P}\left(\beta_j\right) = \prod_{j=1}^{p} \mathrm{P}\left(\mu_j\right) \mathrm{P}\left(A_j\right) \mathrm{P}\left(\phi_j\right) \mathrm{P}\left(\gamma_j\right) \mathrm{P}\left(Q_j | \gamma_j\right) \tag{12}$$

and:

$$\mathrm{P}(\boldsymbol{\theta}) = \prod_{i=1}^{n} \mathrm{P}(\theta_i) \tag{13}$$

No known analytic solution exists to $\mathrm{P}(\boldsymbol{\theta}, \boldsymbol{\beta} | \mathbf{X})$. Moreover, asymptotically exact estimation approaches, such as Markov-chain Monte Carlo and full grid sampling, do not scale well to droplet-based scRNA-seq datasets that can contain thousands (and sometimes tens of thousands) of cells.

For a computationally efficient solution to estimate $P(\boldsymbol{\theta},\boldsymbol{\beta}|\mathbf{X})$, Tempo uses variational inference, an optimization-based approximate Bayesian inference approach. In brief, Tempo uses the list of core clock genes and prior knowledge to initialize an approximate joint posterior distribution $q(\boldsymbol{\theta},\boldsymbol{\beta})$ with differentiable parameters describing its shape. $q(\boldsymbol{\theta},\boldsymbol{\beta})$ is parameterized such that it includes a list of cycling genes and only cycling genes contribute information to the estimate of $\boldsymbol{\theta}$. At initialization, the cycling gene list only includes the user-supplied core clock genes. Tempo optimizes $q(\boldsymbol{\theta},\boldsymbol{\beta})$ to approximate $P(\boldsymbol{\theta},\boldsymbol{\beta}|\mathbf{X})$ by minimizing their KL divergence through an iterative two-step procedure. In Step 1, Tempo estimates the cell phase distributions and gene parameter distributions using only information from the current cycling genes to minimize the KL divergence between the true joint posterior distribution and the approximate joint posterior distribution. In Step 2, Tempo identifies de novo cycling genes whose expression variation is well-described by circadian variation. Approximate gene parameter distributions are computed for current non-cycling genes conditional on the cell phase distributions computed in Step 1 and conditional on the non-cycling genes having non-zero amplitude (i.e., $Q_j$ is set to 1). Tempo then identifies non-cycling genes with expression that is sufficiently better explained by sinusoidal than flat variation and genes with sufficiently high amplitude as de novo cycling genes. De novo cyclers are then added to the cycling gene list. Steps 1 and 2 are repeated, estimating the cell phase posterior distribution from the current cycling genes and identifying de novo cyclers, until the algorithm's stopping criteria are met. The final results of the algorithm are a set of identified cycling genes (the core clock genes and identified de novo cycling genes), and the approximate posterior distributions of all gene and cell parameters. Additional details of the initialization, generative process, and iterative two-step optimization procedure of $q(\boldsymbol{\theta},\boldsymbol{\beta})$ can be viewed in Supplementary Methods 2 and 3.

## Generation of scRNA-seq data with only 24 h sinusoidal components

Realistic gene expression parameters were first estimated from the aorta SMC scRNA-seq dataset. Treating the light-dark sample collection time as the true cell phases, gene posterior distributions were estimated using variational inference according to Tempo's likelihood model. Genes with high amplitudes (Pearson residuals greater than 2 for difference between actual and expected amplitude point estimates) and cycler probability point estimates greater than 0.95 were called as cycling genes, in addition to the annotated core clock genes.

Using the gene parameters estimated from real scRNA-seq, data were simulated according to the following parameters: (1) the number of cells; (2) the mean and standard deviation of log library size of the cells; (3) the number of equally spaced cell phases; (4) the number of flat genes; (5) the number of CCGs. Log library sizes were drawn from a normal distribution. Discrete cell phases were drawn from a multinomial distribution with uniform probabilities. Flat genes and CCGs, and their respective sinusoidal parameters, were sampled with replacement. Given these parameters, scRNA-seq datasets were generated by sampling from our model generative distribution. To simulate scRNA-seq data of unsynchronized cell cultures, cell phases were drawn from 23 equally spaced phases over the circadian cycle. To simulate light-dark cycle time courses, cell phases were drawn from four equally spaced phases over the circadian cycle.

## Generation of scRNA-seq with realistic waveforms

To generate simulated scRNA-seq data with realistic waveforms, we used a bulk RNA array mouse aorta dataset generated by Zhang et al.[23]. This circadian time course sampled tissue every 2 h over 48 h, making it ideal to measure gene waveforms with high fidelity.

JTKCycle[24] was run on these data with a 24 h period, and genes with Benjamini-Hochberg $q$ values <0.05 were considered to be true cyclers; genes with $q$ values greater than 0.05 were considered to be true flat genes. Within each time point, the proportions (i.e., relative abundance) of each gene were computed. For true flat genes, values were fixed to the median value across all time points to produce a flat mean over time. Cell library sizes were drawn from a log10 normal distribution, with mean log10(5000) (i.e., mean library size of 5000 UMIs) and standard deviation 0.5. Counts were then drawn from a Poisson distribution, where the expected value was the gene's proportion multiplied by the library size. In total, 200 cells were simulated for each time point, yielding 4800 simulated cells and 19,065 genes. These simulated data contain ground truths for the cell phases.

As a general measure of temporal signal, for each gene we computed the likelihood ratio test statistic (LRT) of their waveforms over a flat waveform. The distribution of LRTs over all true cyclers can be viewed in Fig. 6a.

To compute the strength of each gene's 24 h sinusoidal component, a Fast Fourier Transform (FFT) was run on the bulk gene proportions at each time point. The strength of each gene's 24 h sinusoidal component was measured as the ratio of the 24 h sinusoidal component's amplitude relative to the sum of all sinusoidal component amplitudes. We refer to this metric as the circadian FFT fraction. The distribution of circadian FFT ratios over all true cyclers can be viewed in Fig. 6b.

## Optimal phase shifting procedure and computation of phase estimation errors

Given the phase estimate of cell $i$ made by method $m$, $\hat{\theta}_i^{(m)}$, we would like to compute its error relative to the true cell phase, $\theta_i$. However, computing the phase error of method estimates is not straightforward. Cyclops, Cyclum, and PCA do not use information about which gene's peak defines the start of the circadian cycle. As such, the absolute latent cell phase estimates of these methods are arbitrary, though the relative ordering of the cells is not. To deal with this, each method $m$'s phase estimates are shifted by $s^{(m)}$ that produces the minimum total error over all cells. $s^{(m)}$ is computed as:

$$s^{(m)} = \operatorname*{argmin}_{s^{(m)*}} \sum_{i=1}^{n} \arccos\left(\cos\left(\hat{\theta}_i^{(m)} - s^{(m)*}\right)\right) \quad (14)$$

The method phase prediction for each cell $i$, $\hat{\theta}_i^{(m)}$, is then shifted:

$$\hat{\theta}_i^{(m:\text{shifted})} = \hat{\theta}_i^{(m)} - s^{(m)} \quad (15)$$

Given the true cells phases, $\theta$, and method-specific shifted phase estimates, $\hat{\theta}^{(m:\text{shifted})}$, we compute the error (in hours) of the phase point estimate made by method $m$ for cell $i$, $\epsilon_i^{(m)}$:

$$\epsilon_i^{(m)} = \frac{12}{\pi} \arccos\left(\cos\left(\theta_i - \hat{\theta}_i^{(m:\text{shifted})}\right)\right) \quad (16)$$

## Simulated data circadian phase estimation

The acrophase priors used by Tempo for the simulated core clock genes were set as follows. First, the prior acrophase location was shifted from the true acrophase value. This was done by drawing a shift from a standard normal distribution, scaling the shift by $2 \times \frac{12}{\pi}$, and then adding the shift to the true acrophase value. Second, the prior acrophase scale of the Von Mises distribution was set such that the width of the 95% interval surrounding the prior acrophase location was 4 h. Tempo, Cyclops, Cyclum, and PCA were run two times: first, considering all genes as input, and, second, restricting the data to the core clock genes.

## Simulated data model stability analysis

For the simulated datasets with results shown in Fig. 2 and Supplementary Figs. 1 2, and 7, Tempo, Cyclops, Cyclum, and PCA were each run five times. As a baseline method to compare against, five random

cell phase predictions were also made for each cell by drawing from a uniform distribution over [0, 2π]. Predictions from Tempo, Cyclops, Cyclum, PCA, and the random method yielded a matrix $\hat{\boldsymbol{\theta}}$, where $\hat{\theta}_{ik}^{(m)}$ denotes the phase point estimate for cell $i$ method $m$ and run $k$. For each cell−method pair, the expected phase across runs for each cell, $E[\hat{\theta}_{ik}^{(m)}]$, was computed using the SciPy[27] circular mean function. The stability (reported in hours) of method $j$'s predictions for cell $i$, $s_i^{(m)}$, was then computed as follows:

$$s_i^{(m)} = \frac{12}{\pi} \times \frac{1}{5} \sum\nolimits_{k=1}^{5} \arccos\left(\cos\left(\hat{\theta}_{ik}^{(m)} - E[\hat{\theta}_{ik}^{(m)}]\right)\right) \qquad (17)$$

Smoothed distributions (Gaussian kernel with bandwidth 0.2 for non-PCA methods, 0.01 for PCA) of this stability metric can viewed in Fig. 2f and Supplementary Figs. 1f, 2f and 7f.

### Aorta data collection and data processing

Eight-week-old C57BL/6J male mice from the Jackson Laboratory were entrained to a 12:12 light-dark cycle for 2 weeks in circadian boxes with ventilation (20–22 °C, RH -50%). At ZT0, ZT6, ZT12, or ZT18 mice were sacrificed (2 per time point). For each mouse, whole aorta was dissected and cut into small pieces (-1 mm), and cells were disassociated with 1.5 ml of enzyme cocktail (DNase−120 U/ml (Worthington, #LS006331), Liberase TM−4 U/ml (Roche, #05401127001), hyaluronidase−60 U/ml (Sigma-Aldrich, #H3506)) in a petri dish at 37 °C for 40 min. A pipette (1 ml) was used to dissociate aortic cells from the tissue pieces every 10 min during the incubation. Cell supernatant was filtered through a 40 μm strainer and washed with RPMI1640 (Gibco, #1187-085) containing 10% fetal bovine serum (FBS, HyClone, #SH30071.03) to inactive the enzyme cocktail. Residual red blood cells were lysed by incubating the cell suspension with Red Blood Cell Lysing Buffer Hybri-Max (Sigma-Aldrich, #R7767) at RT for 1 min. The cells were washed two more times to remove debris with FACS buffer (FBS 2%), EDTA (5 mM, Invitrogen, #15575-038), HEPES (20 mM, Gibco, #15630-080), sodium pyruvate (1 mM, Gibco, #11360-070) in 1x PBS (Gibco, #14190-136), and resuspended in DMEM/F12 (Gibco, #11320-033) media containing 10% FBS for further analysis. Cells from mice sacrificed at the same time point were pooled to form single-cell suspensions (4 suspensions in total). Single cells were then captured and barcoded using the 10X Genomics Chromium platform and sequenced using an Illumina NovaSeq S2 flow cell.

Cell barcode detection, read alignment, and transcript quantification were performed using the 10X Genomics Cell Ranger pipeline. Cells with UMI count less than 1000 and mitochondrial fraction greater than 0.2 were discarded. In addition, cell doublets were detected by the Scrublet program[28] and discarded. A low-dimensional representation for the cells was obtained using UMAP[29] using $z$-score log1p library size normalized counts of the aorta cell type markers from Pan et al. Using the UMAP representation as input, cells were then clustered using the ScanPy[30] implementation of the Leiden algorithm[31]. This yielded five clusters corresponding to vascular SMCs, fibroblasts, macrophages, endothelial cells, and T cells. As only 34 T cells were detected, they were not included in the analyses detailed in this manuscript.

### Light-dark cycle data circadian phase estimation

The acrophase priors used by Tempo for the core clock genes were set as follows. For the aorta cell types, prior acrophase locations were set to the estimated acrophases in bulk liver RNA-seq from Zhang et al. For the hepatocytes from Droin et al.[6], prior acrophase locations were set to the estimated acrophases in bulk liver RNA-seq from Zhang et al. The width of the prior acrophase 95% intervals were set to 2 h for both the aorta and hepatocyte data. *Arntl* was treated as the reference gene.

### Light-dark cycle data out of sample clock likelihood analysis

Out of sample clock likelihoods were computed for each method for the aorta SMCs (5000 train and 13,863 test cells) and aorta fibroblasts (1500 train and 1635 test cells). For a given method $m$ with training set cell phase point estimates $\hat{\theta}_i^{(m:train)}$ and corresponding training set gene parameter point estimates $\boldsymbol{\beta}^{(m:train)}$, we compute the test set core clock expression likelihood $D^{(m)}$ as follows:

$$D^{(m)} = P\left(\mathbf{X}^{(\mathbf{cc:test})} | \boldsymbol{\beta}^{(\mathbf{m:train})}, \hat{\boldsymbol{\theta}}^{(\mathbf{m:test})}\right) \qquad (18)$$

where $\hat{\boldsymbol{\theta}}^{(\mathbf{m:test})}$ are the test set cell phase point estimates from method $m$, and $\mathbf{X}^{(\mathbf{cc:test})}$ is the test set core clock transcript count matrix generated according to:

$$\boldsymbol{\beta}^{(\mathbf{m:train})} = \left(\mu_j^{(m:train)}, A_j^{(m:train)}, \phi_j^{(m:train)}\right) \qquad (19)$$

$$X_{ij}^{(cc:test)} \sim \text{Poisson}\left(\lambda_{ij}^{(m:test)} L_i^{(test)}\right) \qquad (20)$$

$$\log\left(\lambda_{ij}^{(m:test)}\right) = \mu_j^{(m:train)} + A_j^{(m:train)}\cos\left(\hat{\theta}_i^{(m:test)} - \phi_j^{(m:train)}\right) \qquad (21)$$

For Tempo, $\boldsymbol{\beta}^{(\mathbf{m:train})}$ is computed during training. Cyclops, Cyclum, and PCA do not explicitly estimate $\boldsymbol{\beta}^{(\mathbf{m:train})}$ during training, but instead learn a mapping $f(\mathbf{X}^{(\mathbf{train})}, \boldsymbol{\tau}^{(\mathbf{m})}) = \hat{\theta}_i^{(m:train)}$, where $\boldsymbol{\tau}^{(\mathbf{m})}$ represents the learned parameters of method $m$ on the training data. For Cyclops, Cyclum, and PCA, we find the point estimate $\boldsymbol{\beta}^{(\mathbf{m:train})}$ that maximizes the expression likelihood under the following Poisson GLM:

$$X_{ij}^{(train)} \sim \text{Poisson}\left(\lambda_{ij}^{(m:train)} L_i^{(train)}\right) \qquad (22)$$

$$\log\left(\lambda_{ij}^{(m:train)}\right) = \mu_j^{(m:train)} + A_j^{(m:train)}\cos\left(\hat{\theta}_i^{(m:train)} - \phi_j^{(m:train)}\right) \qquad (23)$$

As a lower bound on performance, method out of sample likelihoods were compared to a distribution of likelihoods associated with a random phase inference method, where phases were drawn from a random circular uniform distribution. To build this distribution, $D^{(m)}$ was computed 50 times for the random circular uniform approach. To contextualize how methods performed relative to this random approach, the difference between method log likelihoods and the median of the random log likelihood distribution were computed. These differences were then scaled by the standard deviation of the random log likelihood distribution, and are shown in Fig. 5c and Supplementary Figs. 14–16c.

For additional context, method likelihoods were also compared to likelihoods associated with treating the cell sample collection phases as the cell circadian phases, shown as the "LD" bar in Fig. 5c and Supplementary Figs. 14–16c.

### De novo cycler enrichment analysis

To assess whether de novo cyclers are enriched for a given gene set, we first computed the number of de novo cyclers in the gene set. Empirical null distributions were generated by sampling random gene sets of the same size as the de novo cyclers, and then computing the number of overlaps with the gene set of interest. Empirical $p$ values could then be computed using the empirical null distribution. For both the enrichment of bulk tissue cyclers and cycling genes identified when cell phases were fixed the sample collection phase, the empirical null distributions were generated by sampling 1000 random gene sets.

## Reporting summary

Further information on research design is available in the Nature Research Reporting Summary linked to this article.

## Data availability

Raw sequencing and UMI count matrices for the aorta light-dark cycle time course scRNA-seq data generated were deposited to the Gene Expression Omnibus under accession code "GSE206583". Zipped python AnnData.h5ad objects (containing UMI count matrices, cell collection times etc.) for the aorta SMCs, fibroblasts, endothelial cells, and macrophages are provided as Supplementary Datasets 1–4, respectively. These .h5ad objects can also be viewed at https://www.dropbox.com/sh/tl0ty163vyg265i/AAApt14eybExMMPK7VVDmfvga. The mouse liver hepatocyte data from Droin et al. can be found on GEO with accession code "GSE145197". All other relevant data supporting the key findings of this study are available within the article and its Supplementary Information files or from the corresponding author upon reasonable request. Source data are provided with this paper.

## Software availability

Tempo[32] is provided as an open-source software package available at https://github.com/bauerbach95/tempo. The analyses performed in the manuscript used python version 3.8.5 and Tempo version 0.0.1.dev. Tempo additionally depends on anndata, numpy, pandas, scanpy, sklearn, scipy, statsmodels, tqdm, and pytorch. For the analyses described in the manuscript, Tempo used version 0.7.5. of anndata, 1.19.2 of numpy, 1.2.0 of pandas, 1.6.0 of scanpy, 0.23.2 of sklearn, 1.5.2 of scipy, 0.12.1 of statsmodels, 4.55.1 of tqdm, and 1.9.1 of pytorch. PCA was run using the implementation found in sklearn 0.23.2 and using python 3.8.5. Cyclops and Cyclum were both run using the implementation found in version 0.1 of the Cyclum python package and using python 3.7.9. The aorta data generated were processed using python. For this, python 3.8.5, scrublet 0.2.1, 0.7.5. of anndata, numpy 1.19.2, pandas 1.2.0, scanpy 1.6.0, sklearn 0.23.2, scipy 1.5.2, and statsmodels 0.12.1 were used.

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

## Acknowledgements

This work was supported by 2U54TR001878 (G.A.F.), R01GM125301 (M.L.), and 2T32HG000046-21 (B.J.A.). We thank Sean Anderson, Soon Yew Tang, and Ronan Lordan for generating the scRNA-seq data from mouse aorta, and Christopher J. Adams for a critical reading of the manuscript and testing of the software package.

## Author contributions

This study was conceived of by B.J.A. and led by B.J.A. and M.L. B.J.A. designed the model and algorithm. B.J.A. implemented the Tempo software and led data analysis with input from M.L. and G.A.F. B.J.A. wrote the paper with feedback from M.L. and G.A.F.

## Competing interests

G.A.F. is a Senior Advisor to Calico Laboratories. The remaining authors declare no competing financial interests. All authors declare no competing non-financial interests.
