## [Peer Review File · Nature Communications]

REVIEWER COMMENTS

Reviewer #1 (Remarks to the Author):

Establishing the circadian phase of individual cells in the context of single-cell RNA-sequencing (scRNA-seq) experiments is an important and challenging problem. Toward this end, the authors propose a Bayesian variational inference approach called Tempo.

A few approaches are available to estimate cell phase from scRNA-seq data; Tempo improves upon these by providing better estimation of circadian phase and by quantifying estimation uncertainty, which

is required for downstream analysis and interpretation of results. The approach is well motivated and results on simulated and case study data suggest major improvements over existing methods. Overall, I expect the approach will be of broad interest and widely used. However, below I list a number of questions and concerns.

Major:

Tempo requires a list of core clock genes. Is it not applicable to species for which such genes are not available? Does the list change under different experimental situations, potentially limiting utility within commonly used species?

The authors assume expression follows a sinusoid which seems to be a strong assumption, even for circadian genes which often do not follow a perfect sinusoid. How robust is Tempo to mild/moderate departures in this assumption? While the issue is acknowledged

(lines 354-356), additional attention (e.g. via simulations) is warranted.

Were different normalization methods considered? As above, it would be useful to know how robust Tempo is to various normalization methods and/or to have a recommendation with support for a specific

normalization approach.

Given that Tempo requires core clock genes, it is not clear why it is compared to unsupervised approaches like PCA, as opposed to semi-supervised approaches like Wavecrest or reCAT.

One of the most exciting and high-impact contributions here is the ability of Tempo to identify de novo clock genes, but it seems that this was only illustrated in the simulation study. Were novel genes identified in the case studies?

It is not clear how (or if) Tempo handles the confounding between circadian clock and cell cycle, especially when cell cycle has a similar period length (~24h) as the circadian clock in some cell types. This might be (or might not be) less problematic for time course data where the circadian clock is synchronized by the experiment, but the cell cycle is unsynchronized. However, for single-sample unsynchronized data, both the circadian clock and the cell cycle are unsynchronized. How does Tempo deconvolve the oscillation patterns from the circadian clock vs. cell cycle?

Line 220: "considering all genes as input" was not clear to me. Are the same core clock genes used here?

There is no section on data availability.

Minor:

The authors on reference 13 are not correct.

Reviewer #2 (Remarks to the Author):

Auerbach et al. present a Bayesian approach to temporally order sc-RNAseq measurements taken at different timepoints or from unsynchronized cell cultures in the context of circadian rhythms. The authors address an interesting open question in the field with a powerful new approach that could be used beyond circadian rhythms. I have several questions/concerns and chief amongst those is this: the major contribution of this work is the variational Bayes formulation and solution to this problem, but the emphasis of the manuscript seems to be on its applications to biological data that appears rather superficial with no clear biological insights. Moreover, I wonder given the highly technical/statistically oriented presentation/visualization whether this paper is more suited to a good bioinformatics journal.

Major concerns:

1. One of the motivations for the need for this method is "low information content" of clock genes compared to cell genes. I would like to see support for this claim (analysis or citation). Moreover, if

indeed cell cycle genes are “high information content” and numerous, why does Tempo perform relatively poorly on cell cycle data (compared to circadian datasets)?

2. It is one of my pet peeves that all new method papers show that their method outperforms all others. Objectively, this is probably not true, as it depends on the data used for the comparison and the parameters used. In the same vein, it is unfair to use default parameters of these methods (that are either unsuited to scRNAseq data or when the authors themselves stress that these parameters need to be adapted to each new dataset (see Cyclops) or the simulated data are generated according to assumptions more suited to one method). I would like to see a more balanced discussion of these issues and comparisons in the manuscript. Moreover, on a related point, I fail to see how a method like Cyclum designed to score cell cycle states performs worse than Tempo (in Fig 4), which is designed with circadian data in mind.

3. I also wonder about practical considerations. There are so many “tuning knobs” (controls/choices) for Tempo. How critical is setting these parameters and how does the user go about doing that? In addition, as the authors mention many times, the solutions of Tempo to the ordering are random. In a real scenario, where the ground truth is unknown, what is the selection criteria for a “good” solution? The authors showed only in synthetic data that the solutions from Tempo are highly similar.

4. The authors construct a very detailed likelihood model based on modeling different sources of variability. Is such complexity necessary? In other words, would a simpler model do? I ask this considering the rather disappointing conclusion from real data that only core clock genes were used by Tempo to make the phase prediction (which is what anyone with domain knowledge would suggest).

5. One of the unique features of Tempo is its ability to quantify uncertainty in the phase estimates. Other than the calibration data, which is hard for most people to grasp, it would be nice to see how the uncertainty in the phase estimates of different cells look like? Are they uniform across different phases of the circadian cycle? Are they different for different cell types?

6. The authors first cluster mouse aorta data into different cell types before they run Tempo on each cluster/cell type. I wonder how much the clustering, which is also based on the transcriptome, affects the following Tempo analysis. Some discussion of this issue is probably warranted.

Minor concerns:

1. The authors split the data into training and test sets. What is the logic in deciding the split? (I noticed different proportions in the different cell types).

2. Why do the authors not describe the results for the other cell types from the mouse data in the main manuscript? It would be interesting to see how similar or different they are across cell types.

3. I would like to see a short conceptual description of the different methods compared in the main text for readers to know what the authors mean by PCA or Cyclum or Cyclops.

4. I generally found the text a bit too technical especially if a biological audience is aimed. For e.g., line 541, what does “difference between MAP amplitude and expected MAP amplitude conditional of MAP mesor” mean?

5. I wonder how based in biology is that choice of equally spaced phases for the different cell cycle states? Could this be the source of generally poor results for all methods?

Point-by-Point Response to Reviews

We sincerely thank the reviewers for their constructive comments, which have helped improve the exposition of our manuscript. We have made substantial improvements to our manuscript and highlighted the revised/new parts in red for ease of reviewing. Below are our point-by-point responses to the reviewers' comments. The original reviewers' comments are in **bold italics** and our responses are in normal font colored in purple.

General Note

Please note, in the new version of the manuscript we have taken out the section demonstrating Tempo's application to cell cycle inference. While we firmly believe Tempo's performance should generalize to cell cycle inference, there were several issues related to interpreting method performance on the *Hsiao et al.* and *Buettner et al.* datasets. On the *Hsiao et al.* data, cell phases were highly non-uniformly distributed:

Figure R1: True phase distribution for cells from *Hsiao et al.*

As such, when evaluating the quality of point estimates from a method, the method could achieve artificially low error by predicting a single point for all cells (after the optimal phase shifting procedure described in Method section lines 621 to 638).

On the *Buettner et al.* data, cells were coarsely sorted into 3 cell cycle phases: G1, S, and G2/M. As such, the cell phase labels treated as ground truth still have a large degree of uncertainty. Moreover, it was difficult to decide whether the 3 cell cycle phases should be considered equidistant.

Due to these issues, we decided the paper would be made stronger by cutting out this section and refining the message to solely be about Tempo's application to circadian phase inference.

Reviewer #1 (Remarks to the Author):

Establishing the circadian phase of individual cells in the context of single-cell RNA-sequencing (scRNA-seq) experiments is an important and challenging problem. Toward this end, the authors propose a Bayesian variational inference approach called Tempo. A few approaches are available to estimate cell phase from scRNA-seq data; Tempo improves upon these by providing better estimation of circadian phase and by quantifying estimation uncertainty, which is required for downstream analysis and interpretation of results. The approach is well motivated and results on simulated and case study data suggest major improvements over existing methods. Overall, I expect the approach will be of broad interest and widely used. However, below I list a number of questions and concerns.

Major:

1. Tempo requires a list of core clock genes. Is it not applicable to species for which such genes are not available? Does the list change under different experimental situations, potentially limiting utility within commonly used species?

Tempo is only applicable to situations in which a subset of core clock genes is known beforehand. However, this assumption is not particularly limiting when applying Tempo across experimental situations or species.

Existing bulk and single-cell RNA-seq time course data suggest that the core circadian clock genes oscillate in nearly all tissues and cell types that have been surveyed (with one notable exception being, the gonads (Central and peripheral circadian clocks in mammals. Mohawk JA, Green CB, Takahashi JS. Annu Rev Neurosci. 2012;35:445-62. doi: 10.1146/annurev-neuro-060909-153128. Epub 2012 Apr 5. PMID: 22483041). So while we acknowledge there may be contexts in which the core clock genes do not oscillate, evidence suggests these circumstances are rare. Nevertheless, Tempo implements a statistical test to assess if the core clock genes do not show sufficient cycling signal (either for biological or technical reasons). Tempo estimates cell phase, and then compares the Bayesian evidence (Bayesian analog to the likelihood) of the core clock gene expression to the evidence associated with a random cell phase assignment. If the evidence associated with Tempo's phase estimates do not sufficiently improve upon the evidence from a random phase assignment, this is an indication that there is insufficient cycling signal in the core clock genes. As such, Tempo helps the user diagnose situations in which the user-specified core clock genes do not cycle in a particular context.

Across species, the core circadian clock is strongly conserved, and most observed species demonstrate some form of a functional circadian clock in cells (Intrinsic disorder is an essential characteristic of components in the conserved circadian circuit. Pelham JF, Dunlap JC, Hurley JM. Cell Commun Signal. 2020 Nov 11;18(1):181. doi: 10.1186/s12964-020-00658-y.

PMID: 33176800). Moreover, for species in which the core circadian clock genes are not strongly conserved, there are paralogs that enable functional circadian clocks, such as has been observed in *C. Elegans* (Circadian rhythms identified in *Caenorhabditis elegans* by in vivo long-term monitoring of a bioluminescent reporter. Goya ME, Romanowski A, Caldart CS, Bénard CY, Golombek DA. Proc Natl Acad Sci U S A. 2016 Nov 29;113(48):E7837-E7845. doi: 10.1073/pnas.1605769113. Epub 2016 Nov 14. PMID: 27849618).

2. The authors assume expression follows a sinusoid which seems to be a strong assumption, even for circadian genes which often do not follow a perfect sinusoid. How robust is Tempo to mild/moderate departures in this assumption? While the issue is acknowledged (lines 354-356), additional attention (e.g. via simulations) is warranted.

We thank the reviewer for this insightful recommendation. We found this very helpful for understanding the influence of the sinusoidal assumption on Tempo's results.

To assess the effect of departure from a perfect 24-hour sinusoid on Tempo's performance, we first needed a dataset measuring genome-wide waveforms with high fidelity. For this, we used a bulk aorta RNA microarray time course dataset generated by John Hogenesch's group (<https://www.pnas.org/doi/10.1073/pnas.1408886111>). Samples were collected every 2 hours over 48 hours. As these data were densely sampled, measured 2 full circadian cycles using a bulk RNA method, this dataset was ideal for this purpose. JTKCycle was run on these data with a fixed period of 24-hours, and genes with Benjamini-Hochberg q-values less than 0.05 were considered true positive cyclers.

Using the gene expression waveforms from the bulk dataset, we simulated a scRNA-seq dataset. Within each time point, the proportions, i.e., relative abundances, of each gene were computed. For true flat genes, values were fixed to the median value across all time points to produce a flat mean over time. Cell library sizes were drawn from a log10 normal distribution, with mean log10(5000) (i.e. mean library size of 5000 UMI) and standard deviation 0.5. Gene transcript counts were then drawn from a Poisson distribution, where the expected value was the gene's proportion multiplied by the library size. 200 cells were simulated for each time point, yielding 4800 simulated cells and 19065 genes. These simulated data contain ground truths for the cell phases.

As a general measure of temporal signal, for each gene we computed the likelihood ratio test statistic (LRT) of their waveforms over a flat waveform. The distribution of LRTs over all true cyclers can be viewed in **Figure R2**:

Figure R2: Distribution of true cyclers temporal signal strength, measured according to the likelihood ratio of the true waveform vs. a flat waveform.

To compute the strength of each gene’s 24-hour sinusoidal component, a Fast Fourier Transform (FFT) was run on the bulk gene proportions at each time point. The strength of each gene’s 24-hour sinusoidal component was measured as the ratio of the 24-hour sinusoidal component’s amplitude relative to the sum of all sinusoidal component amplitudes. We refer to this metric as the circadian FFT fraction. The distribution of circadian FFT fractions over all true cyclers can be viewed in **Figure R3:**

Figure R3: Distribution of true cyclers 24-hour sinusoid component strength.

Our analyses suggest the waveforms of the core circadian clock genes are among the most similar to pure 24-hour sinusoids. Next, using this simulated dataset containing ground truths about the cells and genes, we assessed a few different aspects of Tempo’s performance.

First, does using the “true” waveforms of the circadian clock genes (vs. perfect sinusoids) lead to markedly worse performance? In general, we would not expect this to be a significant concern as Tempo performed very well on estimating circadian phases from core clock expression alone in the real datasets evaluated in the manuscript, suggesting circadian clock genes are highly sinusoidal. Indeed, Tempo’s phase point estimates made based on the core clock genes are largely unaffected when using their true waveforms, as errors closely mirror that of the theoretical optimum (**Figure R4**):

Figure R4: Empirical CDF (eCDF) of the error of Tempo’s cell phase point when using only the core circadian clock genes as input. The x-axis is the estimated cell phase error in hours, and the y-axis is the percentage of cells with cell phase errors within a certain number of hours.

Moreover, cell phase uncertainties remain well-calibrated (**Figure R5**).

Figure R5: Uncertainty calibration when run with the core clock genes alone

Second, using these simulations we wanted to assess whether departure from perfect 24-hour sinusoids worsens our ability to call *de novo* cycling genes. Running Tempo with *de novo* cyclers detection, Tempo called 25 *de novo* cycling genes, all of which were true cyclers. The called *de novo* cyclers were among the true cycling genes with the most temporal signal (**Figure R6**). To get a sense of how much temporal signal the called *de novo* cyclers had, we visualized their log₁₀ LRT distribution

Figure R6: Distribution of *de novo* cyclers temporal signal strength

The waveforms of called *de novo* cyclers had modestly stronger 24-hour sinusoidal components than those of all true cyclers (**Figure R7**).

Figure R7: Distribution of the strength of the 24-hour sinusoidal component for genes called as *de novo* cyclers by Tempo

However, the waveforms of called cyclers had notably less pure 24-hour sinusoidal components than those of the core circadian clock genes *Dbp*, *Nr1d1*, and *Arntl*. Undetected cycling genes with similar temporal strength to that of the detected cycling genes demonstrate similarly pure 24-hour sinusoidal components (**Figure R8**).

Figure R8: Bivariate distributions of cycler temporal strength and 24-hour sinusoidal component strength for true cyclers detected or undetected as *de novo* cyclers

This suggests that Tempo's 24-hour sinusoidal component assumption likely does not strongly affect *de novo* cycler detection sensitivity.

Third, we wanted to assess how incorporating the called *de novo* cyclers affected cell phase estimates. On these data, incorporation of *de novo* cyclers improves Tempo's cell phase point estimates (**Figure R9**).

Figure R9: eCDFs of the error of Tempo's cell phase point estimates when run with the core clock genes and *de novo* cyclers

For example, 62% of estimates lie within 3 hours of the true cell phase based on the clock alone; this improves to 72% when incorporating *de novo* cyclers. However, unlike the results on the data simulated with pure 24-hour component sinusoidal waveforms, results on these data suggest Tempo's point estimates incorporating *de novo* cyclers are suboptimal.

Incorporation of *de novo* cyclers yields uncertainty estimates that remain well-calibrated (**Figure R10**).

Figure R10: Uncertainty calibration when run with the core clock genes and *de novo* cyclers

Altogether, these results suggest that Tempo’s assumption of 24-hour component sinusoidal waveforms is reasonable for estimating cell phase from the core circadian clock gene. Moreover, this assumption is reasonable for identifying *de novo* cyclers with high specificity. However, Tempo’s sinusoidal assumption is suboptimal for estimating cell phase when incorporating *de novo* cyclers. This is, perhaps, unsurprising when looking at the distribution of the circadian Fast Fourier Transform (FFT) fraction for the called *de novo* cyclers above. Many called *de novo* cyclers have fractions less than the highly sinusoidal core circadian clock genes *Arntl*, *Dbp*, and *Nr1d1*. Discussion of these results can be viewed in the amended manuscript from lines 226 to 258, and lines 594 to 619.

Despite this limitation, we have demonstrated that Tempo can identify *de novo* cyclers well and improve phase estimates using *de novo* cyclers when waveforms are more strongly sinusoidal. While this may occur infrequently in real data, we have found that Tempo’s computation of core clock Bayesian evidence is an adequate tool to detect when these situations arise. Moreover, in situations when waveforms are not sufficiently sinusoidal, Tempo consistently favors the phase estimates associated with the core clock alone.

Tempo demonstrates a significant improvement over existing tools for estimating phase based on the core clock genes alone. Further, we believe Tempo’s well-calibrated uncertainty quantification will be valuable to researchers. Nevertheless, we believe future strategies to identify *de novo* cyclers and to incorporate their information into phase estimates should be able to handle more flexible waveforms.

3. Were different normalization methods considered? As above, it would be useful to know how robust Tempo is to various normalization methods and/or to have a recommendation with support for a specific normalization approach.

When computing the Negative Binomial likelihood required for Tempo's objective functions, the input counts used are the raw UMI counts. To deal with the impact of library size variation, as many normalization methods aim to do, we include the log library size of each cell as a covariate. As such, and given the literature suggesting that UMI counts are roughly distributed as a Negative Binomial, we felt using the raw counts (adjusting for library size variation) to compute our likelihood was reasonably justifiable.

4. Given that Tempo requires core clock genes, it is not clear why it is compared to unsupervised approaches like PCA, as opposed to semi-supervised approaches like Wavecrest or reCAT.

We intended to compare our method to PCA only as a baseline to improve upon, as PCA was not specifically designed for the problem of interest. We sought to understand how Tempo performed relative to PCA in two different contexts: 1) when PCA was used in a fully unsupervised fashion (i.e. all genes were used as input), and 2) when PCA was used in a semi-supervised fashion, where only the core clock genes were used as input. We have amended the manuscript (lines 179 to 181) to make it clear that PCA was only intended as a baseline.

While we chose to compare against Cyclops, Cyclum, and PCA in the manuscript, there are a number of other existing tools we sought to compare against. We attempted to evaluate reCAT's performance, especially as it is one of the most popular existing tools in this space. Unfortunately, reCAT's approach is very computationally intensive and was not designed to scale to larger droplet-based scRNA-seq datasets containing thousands of cells. As such, we were unable to get reCAT to finish running on the time course light-dark cycle circadian datasets used in the paper to evaluate Tempo. We also attempted to compare to Oscope, a popular phase inference algorithm developed by the same group as Wavecrest and based on a similar principle (extended nearest-insertion algorithm). Unlike Wavecrest, though, Oscope is fully unsupervised. Similar to reCAT, we were unable to get Oscope to successfully finish on the larger circadian scRNA-seq datasets evaluated in the paper. After the recommendation, we did try to run WaveCrest on the aorta fibroblasts (3135 cells), specifying the core circadian clock genes as our marker gene set. This was run on an iMac Pro with 32 GB RAM. Unfortunately, the program did not complete and we received an error message describing the computer ran out of memory:

```

$root
[1] "home"

[1] "~/Desktop/"
<ReactiveValues>
  Values:   AddHeatMap_buttons, AddPlot_buttons, AddTrend_buttons, ConditionVector, DF_buttons, FlipOrder_buttons, Iden_
buttons, InfoFileName, Log_InData, MarkerPlot_buttons, Markers, Meancut, Norm_buttons, Outdir, Outdir-modal, Permu, PlotN
um, Seed, Submit, UseCols, clusterRow, col1, col2, col3, exDynamicPlotFileName, exENINormFileName, exGListFileName, exHea
tMapPlotFileName, exMarkerPlotFileName, exNormFileName, filename, log_whether
  Readonly: TRUE
Warning: Error in : vector memory exhausted (limit reached?)
165: FUN
164: lapply
163: sapply
162: MedianNorm
161: <reactive:In> [/private/var/folders/zs/xw9fd8tj5r107fls0stzgw40000gn/T/Rtmpq384Ez/shinyapp10b7e1a94
716a/WaveCrest-master/server.R#100]
145: In
144: eventReactiveValueFunc [/private/var/folders/zs/xw9fd8tj5r107fls0stzgw40000gn/T/Rtmpq384Ez/shinyapp
10b7e1a94716a/WaveCrest-master/server.R#287]
100: Act
99: renderText [/private/var/folders/zs/xw9fd8tj5r107fls0stzgw40000gn/T/Rtmpq384Ez/shinyapp10b7e1a94716

```

These popular existing tools were developed with smaller plate-based scRNA-seq datasets in mind, and as a result are not well-suited to scale to the larger droplet-based scRNA-seq datasets. While we described this problem in brief in the manuscript, we did not emphasize its importance as strongly as other points. Nevertheless, this is a key consideration as well.

5. One of the most exciting and high-impact contributions here is the ability of Tempo to identify *de novo* clock genes, but it seems that this was only illustrated in the simulation study. Were novel genes identified in the case studies?

Overall, our analyses suggest Tempo reliably identifies *de novo* cycling genes in real data. However, their incorporation does not appear to improve phase estimates in the real datasets we analyzed.

De novo cycling genes were called by Tempo for all real datasets. For the real circadian light-dark cycle datasets 189, 109, 87, 28, and 117 *de novo* cyclers were called for the aorta SMCs, aorta fibroblasts, aorta endothelial cells, aorta macrophages, and liver hepatocytes, respectively. The quality of these cyclers was assessed by two criteria.

First, for each cell type, we ran Step 2 of Tempo assuming the cell phases were equal to their sample collection phase and called *de novo* cyclers. When run in a fully unsupervised manner, Tempo's *de novo* cyclers should be enriched for cyclers that were called when cell phases were fixed to their sample collection phase. Indeed, *de novo* cyclers were enriched for these cyclers for all cell types evaluated (**Figure R11 a – e**).

Second, we expect that *de novo* cyclers detected from cell types are enriched for cycling genes detected from bulk datasets from the same tissue. Using the bulk aorta and liver datasets from Zhang *et al.* (<https://www.pnas.org/doi/10.1073/pnas.1408886111>), genes with JTKCycle q-values less than 0.05 were considered to be true bulk cyclers. Strong enrichment was observed

for the aortic SMCs, aortic fibroblasts, and liver hepatocytes (**Figure R11 f – j**). While more modest enrichment was observed for aortic macrophages (empirical p-value = 0.1165) and no enrichment for aortic endothelial cells (empirical p-value = 0.6759), these cell types compose a smaller proportion of the tissue than SMCs and fibroblasts. As such, we would expect less concordance with the bulk aorta results for these cell types. Altogether, these results suggest Tempo reliably identifies *de novo* cycling genes in real circadian scRNA-seq data.

In contrast to its performance on the simulated data with only 24-hour sinusoidal components, however, Tempo favored cell phase estimates based on the core clock genes alone rather than those including *de novo* cyclers for all real light-dark cycle cell type datasets. As the identified *de novo* cyclers appear to be compelling, it was unexpected that Tempo favored the phase estimates made just based on the core circadian clock genes alone. Our analysis (response to your Major Point 2, above) applying Tempo to simulated data with non-sinusoidal waveforms, suggests our model's sinusoidal assumption may be problematic for incorporation of *de novo* cyclers into phase estimates. While breaking the sinusoidal assumption is out of the scope of this work, we hope this lays the groundwork for future improvements to deal with this potential issue.

We have amended the manuscript to discuss Tempo's ability to identify *de novo* cycling genes. Please view lines 355 to 379 in the amended manuscript.

Figure R11: Tempo called *de novo* cycler enrichment. Enrichment in called cyclers when cell phase set to sample collection times for **a)** aorta SMCs **b)** aorta fibroblasts **c)** aorta endothelial cells **d)** aorta macrophages and **e)** liver hepatocytes. Enrichment in bulk cyclers (JTKCycle q-value less than 0.05) for **f)** aorta SMCs **g)** aorta fibroblasts **h)** aorta endothelial cells **i)** aorta macrophages and **j)** liver hepatocytes

6. It is not clear how (or if) Tempo handles the confounding between circadian clock and cell cycle, especially when cell cycle has a similar period length (~24h) as the circadian clock in some cell types. This might be (or might not be) less problematic for time course data where the circadian clock is synchronized by the experiment, but the cell cycle is unsynchronized. However, for single-sample unsynchronized data, both the circadian clock and the cell cycle are unsynchronized. How does Tempo deconvolve the oscillation patterns from the circadian clock vs. cell cycle?

One benefit of Tempo is that it is robust to confounding due to the presence of cyclical processes other than the process of interest (i.e. confounding due to the cell cycle, when the user is interested in estimating the circadian phase of cells). This is because Tempo requires the user supplies a list of core clock genes, whose temporal variation reflects phase variation in the process of interest (e.g. the circadian cycle).

This is in contrast to existing tools such as Cyclum or Cyclops. These tools solely aim to find a circular projection of the data that maximizes the likelihood of the input genes, and use no other constraints. As such, they are more susceptible to this type of confounding when run without prior feature selection (e.g. without restricting the set of input genes to only the core circadian clock genes).

To demonstrate Tempo is insensitive to cell cycle signals when used for circadian phase inference, we simulated scRNA-seq datasets where gene expression variability is due to circadian and cell cycle variability (in addition to the Negative Binomial error term). We assume 520 genes cycling along the circadian cycle (20 of which are core circadian clock genes), and 300 genes cycling along the cell cycle, and that these gene sets did not overlap. The median library size was 7000 UMIs and we simulated 2000 cells. Cells were simulated to be unsynchronized across the circadian and cell cycles (cell phases were uniformly drawn from 24 equally spaced bins over the circle), and the two types of cell phases were drawn independently.

Running Tempo on these data without *de novo* cycler detection, the MAP cell phases align well with the ground truth circadian phases (AstroPy implementation of the circular correlation coefficient: 0.673). Moreover, Tempo's MAP cell phases appear uncorrelated with the ground truth cell cycle phases (AstroPy implementation of the circular correlation coefficient: 0.030). **Figure R12** shows the empirical CDF's (eCDF's) of Tempo's estimated MAP phases relative to the ground truth circadian and cell cycle phases reiterates that it is not picking up cell cycle signal:

Figure R12: Empirical CDF (eCDF) of cell phase estimate errors for both circadian and cell cycle phase when using the core circadian clock genes as input. The x-axis is the estimated cell phase error in hours, and the y-axis is the percentage of cells with cell phase errors within a certain number of hours.

Running Tempo on these data with *de novo* cycler detection also demonstrates that it is not picking up cell cycle signal. The MAP cell phases align well with the ground truth circadian phases (AstroPy implementation of the circular correlation coefficient: 0.8298). Moreover, Tempo’s MAP cell phases appear uncorrelated with the ground truth cell cycle phases (AstroPy implementation of the circular correlation coefficient: 0.0005). **Figure R13** shows the eCDF’s:

Figure R13: Empirical CDF (eCDF) of cell phase estimate errors for both circadian and cell cycle phase when using all genes as input (i.e. with *de novo* cycler detection). The x-axis is the estimated cell phase error in hours, and the y-axis is the percentage of cells with cell phase errors within a certain number of hours.

7. Line 220: "considering all genes as input" was not clear to me. Are the same core clock genes used here?

We apologize for the lack of clarity. In this analysis, we first ran Tempo on the simulated data without *de novo* cyler detection (i.e. only used the core clock genes to estimate cell phase), and demonstrated that we could indeed estimate cell phase from the core clock genes alone.

We next assessed whether identifying *de novo* cyclers and incorporating them, along with the same core clock genes used previously, improved the cell phase estimates. We refer to the input gene set, in this scenario, as including all genes. We have revised the corresponding text in the revised manuscript.

8. There is no section on data availability.

The data availability section has been amended (lines 854 to 857). To view our aorta data, please view GSE206583 on the Gene Expression Omnibus (reviewer accession token: qvwpcmkoldudbub).

Minor:

1. The authors on reference 13 are not correct.

We thank the reviewer for helping catch our mistake. We have amended reference 13 to contain the correct list of authors and order.

Reviewer #2 (Remarks to the Author):

Auerbach et al. present a Bayesian approach to temporally order sc-RNAseq measurements taken at different timepoints or from unsynchronized cell cultures in the context of circadian rhythms. The authors address an interesting open question in the field with a powerful new approach that could be used beyond circadian rhythms. I have several questions/concerns and chief amongst those is this: the major contribution of this work is the variational Bayes formulation and solution to this problem, but the emphasis of the manuscript seems to be on its applications to biological data that appears rather superficial with no clear biological insights. Moreover, I wonder given the highly technical/statistically oriented presentation/visualization whether this paper is more suited to a good bioinformatics journal.

Thank you for your appreciation of our method being a powerful new approach for circadian rhythms and beyond. Since the main focus of the paper is to present our novel algorithm, the applications to biological data are only to demonstrate the performance of our algorithm when

applied to real data. We believe Nature Communications is an appropriate avenue to publish this work given its broad readership. We hope by publishing our method in a journal like Nature Communications, more biologists will become aware of our method and hence can apply our method to their studies to advance biological discoveries.

Major concerns:

1. One of the motivations for the need for this method is “low information content” of clock genes compared to cell genes. I would like to see support for this claim (analysis or citation). Moreover, if indeed cell cycle genes are “high information content” and numerous, why does Tempo perform relatively poorly on cell cycle data (compared to circadian datasets)?

Indeed, one of the motivations for developing Tempo was the poor performance of existing tools developed for cell cycle when applied to circadian phase inference in scRNA-seq. These approaches make several assumptions that are suboptimal when applied for circadian phase inference.

In hindsight, the comment that circadian cycle genes are enriched for “low information context” relative to the cell cycle was made with inadequate care. The intention was to convey that there are few circadian clock genes that are consistently highly expressed and have high amplitudes (usually only Dbp, Nr1d1, Tef, Hlf, Nfil3, and Arntl consistently satisfy these criteria), and that this is a very difficult problem that requires an approach to deal with that. We thank the reviewer for helping ensure the precision of our language. We have amended the manuscript as such. Please view line 97.

With respect to why Tempo performs worse on cell cycle data (relative to the circadian), we suspect the main reasons are specific to each of the two datasets. With respect to the dataset generated by *Hsiao et al.*, the cells are sampled in a highly non-uniform manner over the cell cycle; the cells are mainly sampled from 2 modes, 0 radians and $\frac{3}{4}\pi$ radians (**Figure R1**). This may lead to worse estimates of the sinusoidal gene expression parameters, and, in turn, worse results for phase estimation.

With respect to the dataset generated by *Buettner et al.*, the cells were coarsely sorted into 3 cell cycle phases: G1, S, and G2/M. As such, the cell phase labels treated as ground truth still have a large degree of uncertainty. Moreover, this dataset contains just under 300 cells, compared to the circadian datasets analyzed containing a few thousand. As such, the parameter estimates made by Tempo are more uncertain relative to the circadian datasets and it may be difficult to obtain a good point estimate.

As explained in the General Note, we have decided to take out the section demonstrating Tempo’s application to cell cycle inference.

Altogether, while Tempo does not demonstrate improved point estimates for the cell cycle datasets analyzed, its point estimates appear to be about as good as the next best performing methods, or modestly worse. More importantly, Tempo provides a well-calibrated uncertainty

quantification that is helpful for result interpretation. We would like to point out that among all existing methods, Tempo is the only method that gives uncertainty quantification.

2. It is one of my pet peeves that all new method papers show that their method outperforms all others. Objectively, this is probably not true, as it depends on the data used for the comparison and the parameters used. In the same vein, it is unfair to use default parameters of these methods (that are either unsuited to scRNAseq data or when the authors themselves stress that these parameters need to be adapted to each new dataset (see Cyclops) or the simulated data are generated according to assumptions more suited to one method). I would like to see a more balanced discussion of these issues and comparisons in the manuscript. Moreover, on a related point, I fail to see how a method like Cyclum designed to score cell cycle states performs worse than Tempo (in Fig 4), which is designed with circadian data in mind.

We agree the aspects you mention can make comparison between methods challenging. To address your concerns, we performed the following investigations.

Interpretation of the results of methods on the simulated data

The simulated data evaluations were primarily motivated by assessing Tempo's performance alone, and sanity checking that it performed as expected. While the generative distribution used for the simulations appears to describe real scRNA-seq data fairly well, we agree caution is required in interpreting these results. More specifically, Tempo's likelihood distribution exactly matches that of the simulation generative distribution, meaning it has an innate advantage relative to competing methods on these data. We agree additional context on this matter would add to the manuscript. Please view lines 262 to 264 for additional discussion.

Despite the general limitations of analyzing method performance on simulated data, Tempo's similar performance between the simulated and real data suggests the simulation generative distribution describes real scRNA-seq data fairly well. For example, the real aorta fibroblasts had 3185 cells, were collected at 4 time points, and had a median library size of 7412 UMI. On these data, 71% of Tempo's point estimates were within 4 hours of the expected cell phases (Supplemental Figure 15a). For comparison, on a sinusoidal simulated dataset with similar characteristics (3000 cells, collected at 4 time points, and a median library size of 20,000 UMI), 75% of Tempo's point estimates were within 4 hours (Supplemental Figure 8). These similar results suggest that the simulations closely recapitulate the generative process of the real data. As such, while the intention of the simulations was primarily to benchmark Tempo, the results of other methods on these data are also likely informative with respect to their performance on real scRNA-seq data.

Competing method hyperparameter selection

We agree that the use of default parameters for competing methods may be suboptimal, depending on the input dataset. Unfortunately, it is not feasible to do an exhaustive grid search

for optimal parameters on each dataset evaluated in the paper. In lieu of this, we used Bayesian Optimization to identify well-performing parameters for both Cyclum and Cyclops. Bayesian Optimization (<https://proceedings.neurips.cc/paper/2012/file/05311655a15b75fab86956663e1819cd-Paper.pdf>) uses a Gaussian process to model the relationship between an algorithm’s hyperparameters (e.g. the learning rate of a neural network) and the quality of its solutions (e.g. the mean error of the phase predictions). Using this Gaussian process, Bayesian Optimization iteratively identifies parameter values where the quality of the solutions is most uncertain, and then evaluates the quality of the solution at those values. This enables Bayesian Optimization to iteratively explore parameter values in a way that is significantly faster than an exhaustive grid search, while at the same time identifying better solutions than a random search. We applied this approach for Cyclum and Cyclops only on the aorta fibroblast dataset. Running Bayesian Optimization for all datasets was not practical, but the aorta fibroblast dataset has moderate cell count (3135 cells) and median library size (7412 UMIs). As such, well-performing parameters identified on this dataset are more likely to generalize to the other datasets. Parameters for Cyclum and Cyclops were identified using 20 Bayesian Optimization steps on these fibroblast data. Bayesian Optimization did suggest more ideal parameter values for Cyclops. The new parameters contained a larger learning rate, more learning epochs, smaller number of principal components to use. Using these new parameters, Cyclops’ median phase error on the aorta fibroblasts improved from 5.487 to 3.975 hours when using the core clock genes only as input. Nonetheless, deeper inspection revealed that the results from the Bayesian Optimization were misleading. Inspecting the cell phase estimate distribution for all fibroblast cells using the new parameters demonstrated phase predictions were essentially the same for all cells (**Figure R14**):

Figure R14: Cyclops cell phase predictions on fibroblasts using “optimal” parameters discovered from Bayesian Optimization.

These predictions artificially achieved a low error after the optimal phase shifting procedure (Method section lines 621 to 638). Given this, we felt the identified parameters for Cyclops did not perform better than the parameters from our initial submission. As such, the results for Cyclops remain the same in the new version of the manuscript.

For Cyclum the best identified parameters from Bayesian Optimization had a median phase error of 4.502 hours on the fibroblasts. This was worse than the fibroblast median phase error from the initial version of the manuscript (4.310 hours), suggesting the initial parameters were closer to optimal. As such, the results for Cyclum in the manuscript remain unchanged.

Cyclum's Performance on Cell Cycle Data

With respect to Cyclum performing worse than Tempo on the *Hsiao et al.* cell cycle data, we too, were surprised. We note that all methods generally perform poorly on the *Hsiao et al.* cell cycle dataset. We speculate this is most likely due to the highly non-uniform phase sampling of cells across the cell cycle, as shown in **Figure R1**.

We spent a significant amount of time inspecting our code to ensure we were running Cyclum correctly. Moreover, as an additional sanity check, we ran Cyclum (and other methods) on the *Buettner et al.* cell cycle data, as the original Cyclum paper benchmarked on this dataset. The *Buettner et al.* dataset contains 288 cells that were sorted into 3 discrete cell cycle states (G0/G1, S, G2/M). We treated these states as equally spaced over the circle (e.g. 0, $\frac{2\pi}{3}$, and $\frac{4\pi}{3}$ radians). In our initial submission, when using all genes as input (as described in the Cyclum paper), 71% of point estimates were within 4 hours of the “true” cell cycle phase (Figure 4c in the original submission). This roughly corresponds to 71% accuracy in assigning cells to the correct discrete cell cycle stage. For comparison, 80% of point estimates were assigned to the correct discrete phases in the original Cyclum paper (Figure 2b of the Cyclum paper: <https://www.nature.com/articles/s41467-020-15295-9>). While the results between our Cyclum run and the original paper's were similar, they were not as close as we anticipated. We believe there are two main reasons for this.

First, the Cyclum paper did not assume the 3 discrete cell phases were equally spaced. The authors classified cells into the 3 stages using a 3 component Gaussian mixture. Moreover, visual inspection of the mixture distribution shows the individual mixture components are not all the same size (e.g. in Figure 2a the S-phase component distribution looks tighter than the other 2 distributions). This suggests that our assumption that the 3 cell cycle stages are equally spaced may partially explain the discrepancy between our results and that of the original Cyclum paper.

Second, our testing of Cyclum on the simulated and real circadian scRNA-seq data suggested its results can be highly unstable (i.e. phase estimates have high variance across any given run). As such, we believe that this may also explain the discrepancy between our results. To assess this, we ran all programs on the *Buettner et al.* cell cycle datasets 10 times. Using all genes as input, we can see that the error eCDF's for Cyclum are highly variable across runs (**Figure R15**):

Figure R15: eCDF of Cyclum’s cell cycle phase estimate errors on the *Buettner et al.* dataset across 10 independent runs.

As such, we believe the discrepancy between our Cyclum results and that of the original Cyclum paper’s can also be explained by the instability of the algorithm.

3. I also wonder about practical considerations. There are so many “tuning knobs” (controls/choices) for Tempo. How critical is setting these parameters and how does the user go about doing that? In addition, as the authors mention many times, the solutions of Tempo to the ordering are random. In a real scenario, where the ground truth is unknown, what is the selection criteria for a “good” solution? The authors showed only in synthetic data that the solutions from Tempo are highly similar.

Thank you for this insightful comment. Below we address your concern on the number of knobs and how they are determined.

We acknowledge there are quite a few parameters as input to Tempo, which enabled us to fine-tune the algorithm during development. However, there are only 5 required parameters (1. the dataset 2. a file path specifying where to write the results out 3. the file path specifying the list of clock genes 4. a file path specifying prior knowledge about the core clock acrophases 5. the name of the reference gene whose peak defines the start of the cycle). None of the required parameters are hyperparameters of the algorithm, and running the algorithm across a wide array of datasets suggests the default values of the hyperparameters generally yield good results. As such, users do not need expertise to select hyperparameter values to get good results.

In broad strokes, the hyperparameter values fall under the following categories and the default values were determined as follows:

- Parameter learning rates
 - It is common to use learning rates for most gradient descent-based procedures anywhere in the range from 10^{-4} to 10^{-1} . It is also common to start with high learning rates at the start of gradient descent, and to gradually lower the learning rate. As such, by default we initialize our parameter learning rates to 10^{-1} . By default, the algorithm lowers the learning rate (by multiplying by the current learning rate by 0.1, by default) whenever the objective function worsens, which is a fairly standard procedure.
- Size of the cell phase grid
 - Enough grid points are needed to be able to capture the shape of the true cell phase posterior distributions. However, the user does not want to use too many grid points, as this increases the computational burden of the algorithm. Another key consideration is the expected resolution of predictions, which should help guide the upper bound on the number of grid points used.
 - By default, we use 24 grid points (1 hour resolution). This is not too computationally burdensome for most datasets (<10k cells). Moreover, we do not believe there is much additional benefit in using more grid points, as it does not seem likely modeling phase differences less than 1 hour would drastically affect results from noisy scRNA-seq data.
 - For larger datasets, we recommend 12 grid points (2 hour resolution) and no fewer than 6 grid points (4 hour resolution).
- Number of samples needed to compute the VI objective
 - More samples lead to less noisy gradient computation for gradient descent. However, more samples lead to larger computation times and more memory usage.
 - By default, we use 3 samples.
 - For larger datasets, we use 1 sample (which is generally standard practice for variational inference). In practice, this does not seem to affect results much.
- Convergence criteria
 - The fraction improvement of the objective function at which point the algorithm halts.
 - Standard practice is often to choose values in the range of 10^{-4} to 10^{-1} . By default, we chose 10^{-3} .
- Threshold for calling highly variable genes
 - Tempo first filters for highly variable genes as candidate *de novo* cyclers (Supplemental Information section 4). These genes are identified based on outlier variances. A model describing the expected mean-variance relationship for all genes is fit. For each gene, the Pearson residual of its amplitude is computed with respect to its expected amplitude.
 - By default, the threshold used to call a gene highly variable is 0.5 (i.e. at least 0.5 standard deviations above the mean).

- The algorithm is largely insensitive to this parameter value. However, larger thresholds can be used to reduce computational burden (i.e. only evaluate most variable genes as potential *de novo* cyclers).

In addition, to help users we have updated the documentation on Github to give more information on recommended values for the optional parameters.

What constitutes a “good” solution:

We believe the ideal parameter estimate will be the one that maximizes the Bayesian evidence (i.e. the average likelihood associated with a set of parameters and their posterior distribution) of core clock gene expression. As such, predictions with larger Bayesian evidence are considered better. As a baseline, we compute the Bayesian evidence associated with a random assignment of cell phases. We then compute the Bayesian evidence associated with our algorithm’s predictions. If the algorithm’s evidence is not sufficiently better than the evidence associated with a random phase assignment, then we alert the user that its results may be unreliable.

While we believe the most important contributions of our method are the significantly improved point estimates and the well-calibrated uncertainty, we also believe our algorithm’s metric of core clock Bayesian evidence is a valuable metric to diagnose good estimates. This is urgently needed, particularly when running other methods whose results we have demonstrated are highly unstable.

Result stability on real data

We previously only demonstrated the stability of Tempo’s results (and the stability of competing tools) on the simulated data. However, given that the real data may have different characteristics, we agree it is valuable to assess model stability on the real data too, and we thank the reviewer for their suggestion. We repeated the model stability analysis for all of the methods on the real circadian light-dark cycle time course datasets. Each method was run 5 times for this stability analysis. Similar to the stability analysis previously performed on simulated data, the circular standard deviation of each cell’s point estimate was calculated across runs, and the distribution over all cells was visualized. We amended the manuscript to show these results (Figure 5d, Supplemental Figures 14-16d, Supplemental Figures 17 – 22c, and lines 327 to 353).

In general, Tempo achieves similar stability in the real circadian data compared to its performance on the simulated data. Most cells have point estimates to within less than an hour between any given run. One notable exception is Tempo’s performance on the endothelial cells. One likely reason for this is because this analysis looks at stability of the point estimate, and the estimates for the endothelial cells are highly uncertain. The distribution of the 90% interval size for the endothelial cells shows most intervals are larger than 10 hours (**Figure R16**):

Figure R16: Distribution of cells' 90% posterior credible interval widths for endothelial cells.

Moreover, looking at the stability of cells with 90% intervals less than 10 hours vs. greater than 10 hours, indeed, shows the least stable cells are also the most uncertain (**Figure R17**):

Figure R17: Left: distribution of estimate stability for cells with 90% credible interval widths less than 10 hours. Right: distribution of estimate stability for cells with 90% credible interval widths greater than 10 hours.

Moreover, we found it surprising that Tempo's point estimates were more unstable on the endothelial cells (EC) than the macrophages, as the datasets have similar numbers of cells (288 EC and 287 macrophages) and median library sizes (6846.5 and 7389 for the ECs and macrophages, respectively). Inspection of the core circadian clock gene expression revealed that stronger expression in the macrophages likely explains their stability differences (**Figure R18**):

Figure R18: Left: each dot represents a core circadian clock gene, and the proportion of cells with non-zero expression in endothelial cells (EC) and macrophages. Center: the mean of transformed expression (library size normalized, log10 transformed with minimum pseudocount of 1e-7) of the clock genes in the endothelial cells and macrophages. Right: the standard deviation of transformed expression (library size normalized, log10 transformed with minimum pseudocount of 1e-7) of the clock genes in the endothelial cells and macrophages.

As we can see, clock expression shows a higher proportion of non-zero expression in macrophages (left subplot of **Figure R18**). Moreover, clock genes have a higher mean and standard deviation in macrophages (center and right subplots of **Figure R18**). This suggests stronger core clock signal in macrophages relative to endothelial cells. This is echoed by the macrophages' larger Bayes factors for Tempo's results relative to random (lines 280 to 287 in the manuscript).

4. The authors construct a very detailed likelihood model based on modeling different sources of variability. Is such complexity necessary? In other words, would a simpler model do? I ask this considering the rather disappointing conclusion from real data that only core clock genes were used by Tempo to make the phase prediction (which is what anyone with domain knowledge would suggest).

In our initial submission, we demonstrate that Tempo can accurately identify *de novo* cycling genes from simulated data assuming perfectly purely 24-hour sinusoidal component waveforms. Moreover, we demonstrated that incorporation of these *de novo* cyclers improves phases estimates, relative to phase estimates made just based on the core clock genes alone.

We demonstrated Tempo reliably identifies *de novo* cyclers for both the real circadian datasets. For a more detailed analysis on this matter, please view our response to Major Point 5 from Reviewer 1 and lines 355 to 379 in the new manuscript.

Nonetheless, while Tempo identifies bonafide *de novo* cyclers in the real data, their incorporation does not appear to improve phase estimates. Based on simulations, we believe the reason for this is Tempo's assumption of strictly sinusoidal waveforms. For a more detailed analysis on this matter, please view our response to Major Point 2 from Reviewer 1 and lines

226 to 259 in the amended manuscript. As such, we believe future efforts should use more flexible approaches to model waveforms that deviate from perfect sinusoids.

Despite this limitation, we have demonstrated that Tempo can identify *de novo* cyclers well and improve phase estimates using *de novo* cyclers when waveforms are more strongly sinusoidal. While this may occur infrequently in real data, we have found that Tempo’s computation of core clock Bayesian evidence is an adequate tool to detect when these situations arise. Moreover, in situations when waveforms are not sufficiently sinusoidal, Tempo consistently favors the phase estimates associated with the core clock alone.

Overall, Tempo demonstrates a significant improvement over existing tools for estimating phase based on the core clock genes alone. Further, we believe Tempo’s well-calibrated uncertainty quantification will be valuable to researchers. While we too were disappointed that the *de novo* cyclers do not improve phase estimates, we believe Tempo’s high specificity in identifying *de novo* cyclers is useful in its own right. Having developed a general framework for phase estimation from a core gene set and *de novo* cycler detection, future efforts will focus on relaxing the sinusoidal assumption in order to maximally utilize *de novo* cyclers for phase estimation.

5. One of the unique features of Tempo is its ability to quantify uncertainty in the phase estimates. Other than the calibration data, which is hard for most people to grasp, it would be nice to see how the uncertainty in the phase estimates of different cells look like? Are they uniform across different phases of the circadian cycle? Are they different for different cell types?

We agree, it is very useful to be able to visualize the cell posterior distributions. Below are some example posterior distributions for individual cells (**Figure R19**).

Figure R19: Example cell phase posterior distributions for cells from real light-dark cycle circadian datasets.

In general, most cell posterior distributions are unimodal and have shapes similar to circular normal distributions. However, situations arise in which posterior distributions are unimodal but deviate from circular normality (e.g. liver hepatocyte cell 4 in the examples above). Situations also arise in which there is ambiguity and the cell posterior is bimodal (e.g. aorta fibroblast cell 4 in the examples above). This likely occurs in situations for which only expression is observed in genes with similar peak times (e.g. if *Dbp* and *Nr1d1* both peak around CT12, and they are the only genes with non-zero expression).

To help our readers get a general sense of posterior distribution shapes across time points, we included a visualization of the average cell posterior distribution for Tempo (and point estimate densities for other methods) for the aorta SMCs in Figure 4 in the main text of the amended manuscript. The same visualization for other cell types of the average cell posterior distributions can be viewed in Supplementary Figures 9-13.

Across cell types, there are general differences. These can be seen in Supplementary Figures 10-14. Moreover, for each cell type, we look at the distribution of the 90% posterior interval widths (a measure of uncertainty) in **Figure R20**:

Figure R20: Histograms showing distributions of cell uncertainties (measured by the size of the 90% posterior credible intervals) for all real circadian datasets analyzed.

In general, SMCs estimates are the most certain, while estimates for liver hepatocytes are by far the most uncertain. Differences in certainty across cell types, in this case, is largely explained by differences in library sizes across cells: the SMCs, fibroblasts, endothelial cells, and macrophages have median library sizes of 13646, 7412, 6846.5, and 7389 UMIs, respectively. In

contrast, the hepatocytes have a median library size of only 1965 UMIs. As such, the hepatocyte phase estimates are going to be much more uncertain.

Across time points, ZT0 and ZT12 time points tend to be slightly more certain than ZT6 and ZT18 time points. This is because most core clock genes (and most circadian oscillators, in general) peak at ZT0 or ZT12 (dawn or dusk). As such, there is more ambiguity at intermediate time points.

6. The authors first cluster mouse aorta data into different cell types before they run Tempo on each cluster/cell type. I wonder how much the clustering, which is also based on the transcriptome, affects the following Tempo analysis. Some discussion of this issue is probably warranted.

We thank the reviewer for raising this key point. Ideally all cells given as input to Tempo share the same sinusoidal gene expression parameters. This may not always be the case. For example, in a large cluster of cells there may exist a subcluster that has higher amplitude for a gene relative to the rest of the cells. Tempo's Bayesian approach is well-suited to handle this situation. In such a situation, the gene posterior distribution will widen and contain additional variance. And as a consequence, the cell posterior distributions will also contain additional variance and uncertainty. So while it is ideal to feed cells with homogeneous sinusoidal gene expression parameters as input to Tempo (as the phase estimates will contain less certainty), Tempo is equipped to deal with this problem.

While previously we noted this assumption in the methods section (see lines 481 to 482), we feel it is important to describe this in the main body of the manuscript as well. Please see line 125 for this addition.

Moreover, we have added additional discussion of the matter and a proposed future solution of the issue in the discussion section. Please view lines 420 to 425.

Minor concerns:

1. The authors split the data into training and test sets. What is the logic in deciding the split? (I noticed different proportions in the different cell types).

We performed the out-of-sample analysis (out-of-sample core clock gene likelihood) for only the smooth muscle cells (SMCs) and fibroblasts. We decided only the SMCs and fibroblasts had a sufficient number of cells to yield confident parameter estimates in this analysis. The main reason the SMCs and fibroblasts have differing train/test proportions is that the analysis was intended to only compare how methods perform against each other within each cell type – not to compare across cell types.

Admittedly the train/test split for the two cell types was heuristic. Based on existing datasets, most clock genes have pseudobulk proportions of at least 10^{-6} at their lowest point of expression and at most 10^{-4} at the peak of their expression. The median library size was at least

7k for the aorta cell types. We felt 1500 cells for the fibroblast training set would be sufficient to yield relatively confident parameter estimates, since this number of cells yielded a non-zero expected number of UMI detected in pseudobulk at the bottom of expression (1500 cells / 4 time points * 7000 UMIs * 10^{-6} = 2.625 expected UMI in pseudobulk at the bottom of expression for clock genes). Moreover, we felt 1500 test set cells would be an adequate amount to achieve a stable estimate of the out-of-sample clock gene likelihood.

As we had a plethora of SMCs (~18k), we decided to use a larger training set to get more confident estimates as we could.

2. Why do the authors not describe the results for the other cell types from the mouse data in the main manuscript? It would be interesting to see how similar or different they are across cell types.

While we agree it would be ideal to include the results of the other cell types for the circadian data in Figure 5 of the main text (which just shows the results for the SMCs), unfortunately this was not practical due to space constraints for individual figures. While we considered adding additional figures for each individual cell type (similar to Figure 5), we ultimately decided against this. We felt that the message across the cell types is largely similar and doing so may have been repetitive for readers. So, we included the results for the other cell types in Supplementary Figures 9 - 22. However, we did feel it was important to give readers a quick summary of how the methods performed across all cell types. As such, in the main text we include Table 1 to help summarize this information.

3. I would like to see a short conceptual description of the different methods compared in the main text for readers to know what the authors mean by PCA or Cyclum or Cyclops.

We thank the reviewer for their suggestion, and have amended the manuscript as such. Please view lines 176 to 183 for the addition.

4. I generally found the text a bit too technical especially if a biological audience is aimed. For e.g., line 541, what does “difference between MAP amplitude and expected MAP amplitude conditional of MAP mesor” mean?

We agree that the concept pointed out by the reviewer may not be clear to readers. Prior to submission, we grappled with how to best convey this concept, though admittedly we did not come to a satisfying result.

To explain what we mean, as one criterion to identify *de novo* cyclers, we wanted to identify genes with high outlier amplitudes. However, it should be noted that genes with low mesors generally have high estimated amplitudes while genes with high mesors have low estimated

amplitudes. As such, when assessing if a gene has high amplitude, we'd like to adjust for its mesor.

To do this, we first needed to come up with an expectation of a gene's amplitude given its mesor. So for every gene, we took Tempo's maximum *a posteriori* (MAP) estimate of the mesor and amplitude. We then used a nonparametric regression model to fit the relationship between the mesor and amplitude. Using this model, we could obtain the expected amplitude of a gene given its mesor.

Using this nonparametric model, we then could compute a gene's expected MAP amplitude given its MAP mesor. We then computed the difference between the gene's actual MAP amplitude and this expectation (which were then scaled by the standard deviation of this model). Positive differences indicated that the gene's amplitude was larger than expected.

We hope this provides some clarity about the line the reviewer pointed out. To help our readers, we decided to simplify this line and have amended the manuscript as such. Please view the changes on lines 579 to 580.

5. I wonder how based in biology is that choice of equally spaced phases for the different cell cycle states? Could this be the source of generally poor results for all methods?

We agree that treating G1, G2/M, and S cell cycle states as equally spaced (for the *Buettner et al.* dataset) may be suboptimal. However, this decision was made as it felt like the simplest approach and that other alternatives were non-obvious. We agree that it could be a source of poor results on both cell cycle datasets. We also believe there are other factors that explain the poor results (as touched upon in our response to point 1 by the reviewer): 1) noisy cell label annotations 2) highly non-uniform sampling across the cell cycle (for the *Hsiao et al.* dataset) 3) few cells (relative to the circadian datasets).

As explained in the General Note, we have decided to take out the section demonstrating Tempo's application to cell cycle inference.

REVIEWERS' COMMENTS

Reviewer #1 (Remarks to the Author):

The authors have carefully and comprehensively addressed all of my comments. Congratulations on this nice work!

Reviewer #2 (Remarks to the Author):

I thank the reviewer for their efforts to address my comments. I am satisfied with their response. Good Luck